

# High-resolution Holocene record from Torfdalsvatn, north Iceland, reveals natural and anthropogenic impacts on terrestrial and aquatic environments

David J. Harning[1,2], Christopher R. Florian[1,2,3], Áslaug Geirsdóttir[2], Thor Thordarson[2], Gifford H. Miller[1],
Yarrow Axford[4], Sædís Ólafsdóttir[5]

[1]Institute of Arctic and Alpine Research, University of Colorado, Boulder, CO
[2]Faculty of Earth Sciences, University of Iceland, Reykjavík, Iceland
[3]National Ecological Observatory Network, Battelle, Boulder, CO
[4]Department of Earth and Planetary Sciences, Northwestern University, Evanston, IL
[5]Reykjavík Energy, Reykjavík, Iceland

*Correspondence to*: David J. Harning (david.harning@colorado.edu)

**Abstract.** Open questions remain around the Holocene variability of climate in Iceland, including the relative impacts of natural and anthropogenic factors on Late Holocene vegetation change and soil erosion. The lacustrine sediment record from Torfdalsvatn, north Iceland, is the longest known in Iceland (≤12000 cal a BP) and along with its high sedimentation rate, provides an opportunity to develop high-resolution quantitative records that address these challenges. In this study, we use two sediment cores from Torfdalsvatn to construct a high-resolution age model derived from marker tephra layers, paleomagnetic secular variation, and radiocarbon. We then apply this robust age constraint to support a complete tephrochronology (>2200 grains analyzed in 33 tephra horizons) and sub-centennial geochemical (MS, TOC, C/N, $\delta^{13}$C, and BSi) and algal pigment records. Along with previously published proxy records from the same lake, these records demonstrate generally stable terrestrial and aquatic conditions during the Early and Middle Holocene, except for punctuated disturbances linked to major tephra fall events. During the Late Holocene, there is strong evidence for naturally driven algal productivity decline beginning around 1800 cal a BP. These changes closely follow regional Late Holocene cooling driven by decreases in Northern Hemisphere summer insolation and the expansion of sea-ice laden Polar Water around Iceland. Then at 880 cal a BP, ~200 years after the presumed time of human settlement, a second shift in the record begins and is characterized by a strong uptick in landscape instability and possibly soil erosion. Collectively, the Torfdalsvatn record highlights the resilience of low-elevation, low-relief catchments to the pre-settlement soil erosion in Iceland, despite a steadily cooling background climate. The precisely dated, high-resolution tephra and paleoenvironmental record from this site can serve as a regional template for north Iceland.



## 1 Introduction

As the planet continues to warm, paleoclimate information is increasingly more important to constrain Earth system models (Tierney et al., 2020). Within the northern North Atlantic, Iceland serves as an ideal location as the island sits at the confluence of major atmospheric and oceanic circulation patterns integral to global heat distribution (Wunsch, 1980; Marshall et al., 2001). Continuous and high-resolution lake sediment records from Iceland form the backbone of its recent geologic climate history. These empirical records have analyzed physical properties for glacier history (Larsen et al. 2011; Striberger et al., 2012;

Harning et al., 2016a, 2016b), pollen, macrofossils, and sedimentary ancient DNA (*sed*aDNA) for plant community (Rundgren, 1995, 1998; Hallsdóttir and Caseldine, 2005; Gathorne-Hardy et al., 2009; Eddudóttir et al., 2015, 2016; Alsos et al., 2021; Geirsdóttir et al., 2022; Harning et al., 2023), bulk geochemistry for soil erosion and diatom productivity (Geirsdóttir et al., 2009, 2013, 2019, 2020, 2022; Larsen et al., 2012; Blair et al., 2015; Harning et al., 2018a; Tinganelli et al., 2018; Bates et al., 2021), lipid biomarkers for fire activity (Ardenghi et al., 2024), and chironomids and lipid biomarkers for quantitative

temperature (Caseldine et al., 2003; Axford et al., 2007, 2009; Langdon et al., 2010; Holmes et al., 2013; Harning et al., 2020; Richter et al., 2020). The Holocene Thermal Maximum (7900 to 5500 cal a BP, Caseldine et al., 2006; Geirsdóttir et al., 2013) has been a particular focus as it provides a potential analogue for future environmental change – current estimates suggest that summer temperatures were ~3 ºC warmer than present (Flowers et al., 2008; Harning et al., 2020). As such, these proxy records have then been used to test and validate glacier models (Flowers et al., 2008; Anderson et al., 2018) and project future

environmental scenarios (Anderson et al., 2019).

Research on Iceland's Holocene paleoclimate has also centered around the impact of human settlement during the last millennium. The classic paradigm is that following the settlement of Iceland (i.e., *Landnám,* ~870 CE, Vésteinsson and McGovern, 2012), soil erosion was intiated due to deforestation and overgrazing by newly introduced livestock (e.g., Thórarinsson, 1944, 1961; Dugmore and Buckland, 1991; Dugmore and Erskine, 1994; Hallsdóttir, 1995) with climate and

volcanic forcings playing a secondary role (Thórarinsson, 1961; Gerrard, 1991). While human impact certainly affected the Icelandic landscape, empirical evidence demonstrates that the natural decline of woody vegetation (Streeter et al., 2015) and increased and sustained soil erosion began several centuries prior to known human presence (Geirsdóttir et al., 2009, 2020). These natural changes have been ascribed to Late Holocene cooling, due to diminishing Northern Hemisphere summer insolation and the southward migration of the Polar Front and sea ice towards Iceland (e.g., Axford et al., 2007; Geirsdóttir et

al., 2013; Cabedo-Sanz et al., 2016; Harning et al., 2021), as well as volcanic eruptions whose tephra deposits initiate erosion through the abrasion and destruction of vegetation that stabilizes the underlying soil (Larsen et al., 2012; Geirsdóttir et al., 2013; Blair et al., 2015; Eddudóttir et al., 2017). Icelandic lake sediment studies provide optimal archives to explore the relationship and phasing of these various mechanisms due to the lake's continuous sedimentation and high sedimentation rates that afford sub-centennial records of environment and climate variability and robust age control derived from geologically

instantaneous tephra markers (e.g., Thórarinsson, 1944; Larsen and Eiríksson, 2008), among other geochronological techniques (i.e., radioactive isotopes and paleomagnetic secular variations, PSV, Ólafsdóttir et al., 2013).





In this study, we revisit the sediment of Torfdalsvatn, a lake in North Iceland (Fig. 1). Due to it being the only known non-marine sedimentary record in Iceland that extends to ≤12000 cal a BP, its lacustrine record has attracted considerable attention (e.g., Björck et al., 1992; Rundgren, 1995, 1998; Rundgren and Ingólfsson, 1999; Florian, 2016; Alsos et al., 2021; Harning et al., 2024). Important contributions include the major elemental composition of regional marker tephra layers (Björck et al., 1992; Alsos et al., 2021), which for the Early Holocene section have been recently revised (Harning et al., 2024). In addition, several paleoenvironmental records have been developed using pollen, macrofossils, chironomids, and plant *sed*aDNA to explore questions related to the postglacial colonization of plants and Holocene temperature variability (Björck et al., 1992; Rundgren, 1995, 1998; Axford et al., 2007; Alsos et al., 2021). Here, we first use two lake sediment cores from Torfdalsvatn to construct a high-resolution age model derived from marker tephra layers, paleomagnetic secular variation, and radiocarbon. We then apply this robust age constraint to support a complete tephrochronology (>2200 grains analyzed in 33 tephra horizons) and sub-centennial geochemical (MS, TOC, C/N, $\delta^{13}C$, and BSi) and algal pigment proxy records. Collectively, and along with previously published proxy records from Torfdalsvatn, these new terrestrial and aquatic records provide key insight into the paleoenvironmental history of North Iceland during the Holocene.

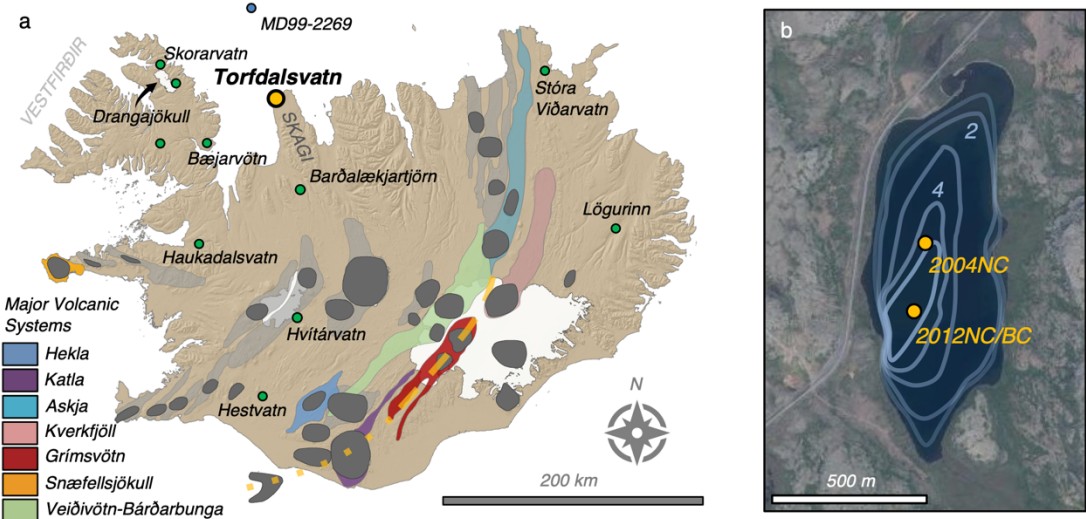

**Figure 1: Overview map of Iceland. a) Locations of major volcanic systems (dark gray denotes central volcano), and terrestrial (green) and marine sites (blue) mentioned in the text. b) Close-up of Torfdalsvatn, its bathymetry (1-m isolines), and location of lake sediment core sites for 2012NC (this study), 2012 BC (Harning et al., 2024), and 2004NC (Axford et al., 2007). Base aerial imagery courtesy of Loftmyndir ehf.**

## 2 Methods

### 2.1 Study site and sediment core collection

Torfdalsvatn (66.06°N, 20.38°W) is a relatively small (0.4 km²), shallow (5.8 m depth) lake located at 52 m asl on the northwest coast of the Skagi peninsula with a catchment area of 2.76 km² (Fig. 1). The surrounding bedrock is composed of Pliocene-



Pleistocene age (3.3 to 0.8 Ma) basaltic lava successions separated by thin sedimentary units (Harðarson et al., 2008). Soils within the low-relief lake catchment are thin and sparse, but where present, are composed of brown to histic andosols as well as some peat (Arnalds and Gretarsson, 2001). The modern vegetation is broadly characterized as moss heathland.

In February 2012, we recovered lake sediment cores from the lake's depocenter using a Nesje piston coring system atop a lake-ice platform. We captured ~8.4 m of continuous sediment in two successive drives (TORF12-2A-1N and TORF12-
2A-2N; referred to as 2012NC) reaching dense, deglacial sediment at the base (Fig. 1b). The cores were split and measured for density and magnetic susceptibility on a GEOTEK (MSCL-S) core scanner at the University of Iceland. Here, we also present new tephra compositional data from a 5.4 m-long sediment record collected from a shallower water depth in February 2004 (04-TORF-01, i.e., 2004NC, Fig. 1b) previously studied for chironomid assemblages (Axford et al., 2007).

**2.2 Tephra sampling and compositional analysis**

Tephra layers from 2012NC and 2004NC were located by visual inspection (or high MS values for those that are not visible, i.e., cryptotephra) (Fig. 2) and sampled along the vertical axis. For the ~26 cm thick G10ka Series unit in 2012NC, samples were taken every 1 cm. Each sample was sieved to isolate glass fragments between 125 and 500 µm and embedded in epoxy plugs. Tephra glass from 2012NC along with some layers from 2004NC were analyzed at the University of Iceland on a JEOL JXA-8230 election microprobe using an acceleration voltage of 15 kV, beam current of 10 nA and beam diameter of 10 µm
(see Supporting Data). The international A99 standard was used to monitor for instrumental drift and maintain consistency between measurements (Table 1 and Supporting Data).

Using the compositional dataset, tephra layer origins were then assessed following the systematic procedures outlined in Jennings et al. (2014) and Harning et al. (2018b). Briefly, based on $SiO_2$ wt% vs total alkali ($Na_2O+K_2O$) wt%, we determine whether the tephra volcanic source is mafic (tholeiitic or alkalic), intermediate and/or rhyolitic. From here, we objectively
discriminate the source volcanic system through a detailed series of bi-elemental plots produced from available compositional data on Icelandic tephra (see Supporting Data). Merging the volcanic system source with relevant stratigraphic and chronologic information permits the identification of the source eruption if the eruption is previously known. If a tephra horizon conforms to the composition of tephra of known age, then it forms a fixed point in the age model (i.e., marker tephra layer). If this is not possible, then an age is extracted from the age model (see Section 2.5) and named according to its volcanic source and modeled
time of deposition. As an example, the Katla tephra layer identified at 142.5 cm depth (1270 cal a BP) is termed "Katla 1270". However, tephra layers from the historical period are classically labeled as age CE, which we specify where needed.



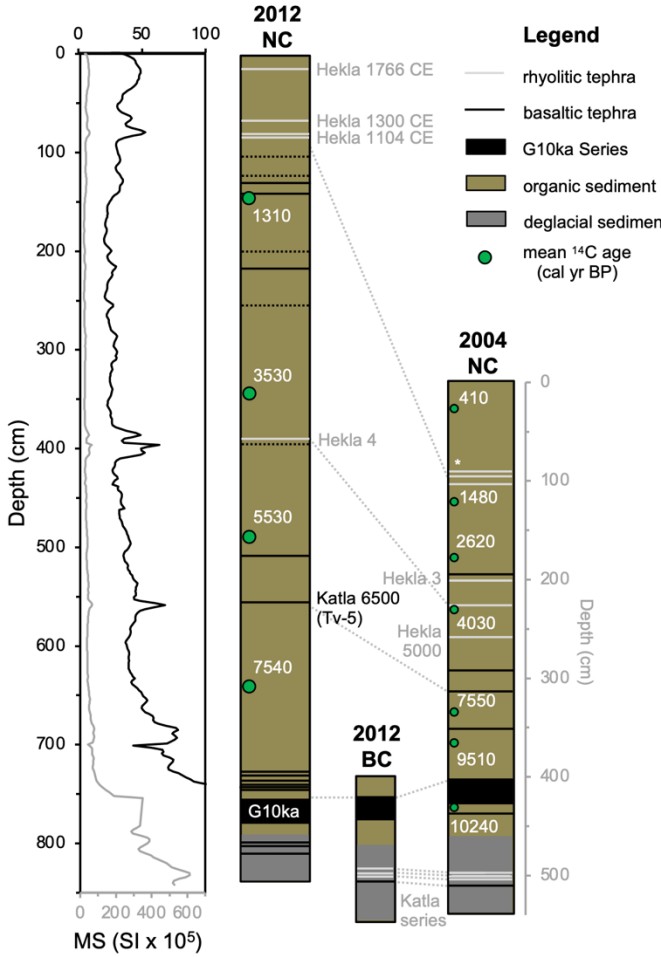

**Figure 2:** Physical and lithological characteristics of Torfdalsvatn lake sediment. Magnetic susceptibility (MS, black and gray are two different scales) record and simplified lithostratigraphy shown for core 2012NC (this study), and lithostratigraphy for cores 2012BC (Harning et al., 2024) and 2004NC (Axford et al., 2007). Note that the 2004NC log is plotted on its own depth axis (gray). See Fig. 8 for stratigraphic detail of the Hekla 1104 CE tephra layers in core 2004NC. Key marker tephra are labeled in each core, and tephra layer correlations between records are highlighted with dashed gray lines. Depth and mean calibrated ¹⁴C ages of macrofossils samples shown for 2012NC and 2004NC with green circles. See Table 1 for complete radiocarbon information.

**Table 1:** Lake sediment radiocarbon information used in this study. All calibrated ages use the most recent radiocarbon calibration curve (IntCal20, Reimer et al., 2020).

| Lab ID | Core name | Depth (cm) | Material | Conventional ¹⁴C age ± σ | Calibrated age BP ± σ | Reference |
|---|---|---|---|---|---|---|
| CURL-15806 | 2012NC | 148 | Aquatic macrofossil | 1390 ± 15 | 1300 ± 10 | This study |
| CURL-15814 | 2012NC | 346 | Aquatic macrofossil | 3305 ± 15 | 3520 ± 40 | This study |



| CURL-15812 | 2012NC | 489.5 | Aquatic macrofossil | 4785 ± 20 | 5530 ± 50 | This study |
| CURL-15794 | 2012NC | 642 | Aquatic macrofossil | 6660 ± 20 | 7540 ± 30 | This study |
| AA60639 | 2004NC | 27.5 | Plant macrofossil | 370 ± 40 | 410 ± 80 | Axford et al. (2007) |
| NSRL-14520 | 2004NC | 120 | Plant macrofossil | 1635 ± 20 | 1480 ± 60 | Axford et al. (2007) |
| NSRL-14765 | 2004NC | 180 | Humic acid | 2515 ± 20 | 2620 ± 100 | Axford et al. (2007) |
| NSRL-14517 | 2004NC | 227.5 | Humic acid | 3685 ± 15 | 4030 ± 50 | Axford et al. (2007) |
| NSRL-14519 | 2004NC | 337.5 | *Betula* leaf | 6690 ± 20 | 7550 ± 30 | Axford et al. (2007) |
| NSRL-14766 | 2004NC | 369 | Humic acid | 8505 ± 15 | 9510 ± 20 | Axford et al. (2007) |
| NSRL-14518 | 2004NC | 432 | Humic acid | 9100 ± 25 | 10240 ± 10 | Axford et al. (2007) |

## 2.3 Paleomagnetic secular variation

Continuous u-tube channels of sediment were taken from the center of core 2012NC and measured for paleomagnetic secular
variation (PSV) following the methods of Ólafsdóttir et al. (2013). Torfdalsvatn's distinct characteristic remanent
magnetization (ChRM) declination and inclination features were tuned to the master GREENICE PSV stack through 14 tie
points (Fig. 3a-b). The GREENICE chronology is based on 47 [14]C dates derived from two PSV-synchronized marine sediment
cores (MD99-2269 and MD99-2322, Stoner et al., 2007, 2013), in addition to eight marker tephra of known age (Kristjánsdóttir
et al., 2007).

## 130 2.4 Radiocarbon

2012NC was inspected for aquatic plant macrofossils to provide additional chronological control. Terrestrial macrofossils were
avoided as it has been shown that their [14]C ages are typically too old for their stratigraphic position in Icelandic lake sediments
(Sveinbjörnsdóttir et al., 1998; Geirsdóttir et al., 2009). The four aquatic macrofossil fragments picked from 2012NC (Fig. 2)
were gently rinsed with deionized water to remove sediment and freeze-dried. Samples were given an acid-base-acid
pretreatment and graphitized at the University of Colorado Boulder, then measured by AMS at the University of California
Irvine. Conventional [14]C ages were calibrated using OxCal 6.0 (Bronk Ramsey, 2009) and the IntCal20 calibration curve
(Reimer et al., 2020), and are reported in cal a BP (Table 1).



## 2.5 Chronology and age model

The 2012NC age model relies on a combination of key marker tephra layers whose ages are known from from the historical
period (*n=2*, Hekla 1766 and Hekla 1300, e.g., Thórarinsson, 1967), PSV tiepoints (*n=14*), and [14]C-dated macrofossils (*n=4*).
An age model was generated using the open-source R package rbacon (Blaauw and Christen, 2011; R Core Team, 2021), the
IntCal20 calibration curve for the [14]C samples (Reimer et al., 2020), and an uncertainty of ± 50 cal a BP for each PSV tie point
(e.g., Ólafsdóttir et al., 2013) (Fig. 3c). Non-marker tephra layer ages are derived from the mean value of model iterations (red
line) and the 95% uncertainty envelope (grey lines) (Fig. 3c).

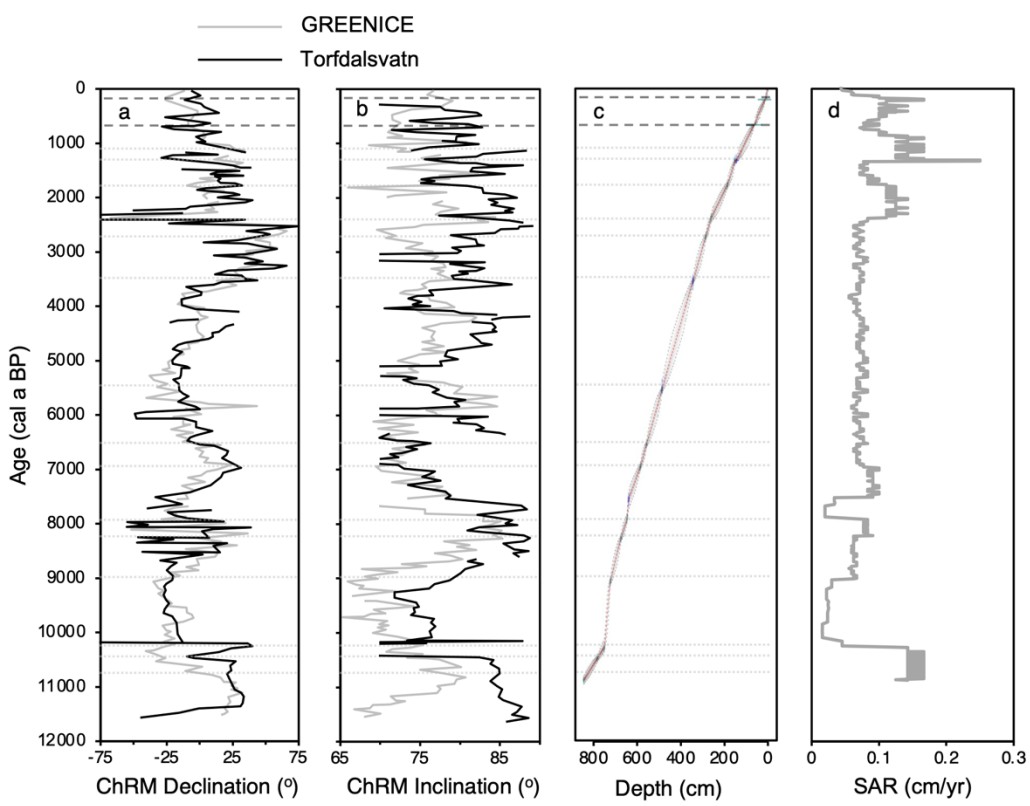

**Figure 3: Age model results. PSV data from core 2012NC (black) compared to the master GREENICE record (gray) in terms of a)
declination and b) inclination. Marker tephra layers used in the PSV synchronization are shown with dark gray dashed lines and
PSV tie points used in the age model are shown with light gray solid lines. c) rbacon age model constructed from the PSV tie points
(*n* = 14), marker tephra layers (*n* = 2), and radiocarbon ages of plant macrofossils (*n* = 4) (Blaauw and Christen, 2011; R Core Team,
2021), where solid red line reflects the median of model iterations, and the outer gray lines reflect the 95% confidence envelope. d)
Sedimentation accumulation rates (SAR, cm/yr) derived from the rbacon age model.**

## 2.6 Bulk geochemistry

Total carbon (TC), total nitrogen (TN), and $\delta^{13}C$ (relative to VPDB) were measured on 407 samples using a CE NC 2500
elemental analyzer interfaced to a Finnigan Delta V isotope ratio mass spectrometer at the Carnegie Geophysical Laboratory.



Isotopic ratios were corrected for drift using an acetanilide standard, and duplicate analyses show a precision of less than 0.2 ‰ for each isotope ratio. For a subset of these samples ($n$=281), we also measured biogenic silica (BSi) by Diffuse Reflectance Fourier Transform Infrared Spectrometry (FTIRS) on a Bruker Vertex 70 with a Praying Mantis diffuse reflectivity accessory (Harrick) at the University of Colorado Boulder. We report values in FTIRS - Fourier Transform Infrared Spectroscopy absorbance units (e.g., Harning et al., 2018a).

## 2.7 Algal pigments

Algal pigments were measured at the University of Colorado Boulder following the methodology described in Leavitt and Hodgson (2001) to minimize pigment degradation. The archive half of sediment core was used for pigment analysis because it was sealed, stored cold, and not visibly oxidized at the time of sampling and all subsamples were kept frozen under nitrogen until the time of measurement. Briefly, following core splitting, algal pigments were immediately solvent extracted from 246 freeze dried sediment samples using 6 mL of 80:20 mixture of acetone:methanol overnight in amber vials under $N^2$ at -10 °C. Samples were touch mixed and sonicated to disperse sediment and increase extraction efficiency, then filtered through a 0.2 µm PTFE syringe filter which was then rinsed with 2 ml of acetone to recover sample residue from the filter. The extract was dried down and rehydrated with a known volume of acetone containing a known concentration of α-tocopherol standard. Samples were placed in the refrigerated autosampler of an Agilent 1200 series HPLC and derivatized to improve chromatographic behavior immediately prior to injection with an equal volume of 28 mM tetrabutyl ammonium acetate in water. A binary mobile phase system was used with solvent A composed of a 70:30 mixture of methanol:28 mM tetrabutyl ammonium acetate in water and solvent B composed of pure methanol at a flow rate of 1 mL min$^{-1}$. An Agilent Eclipse XDB-C18 4.6 x 150 mm column was used to separate pigments whose absorbance of visible light was detected by an Agilent Diode Array Detector (DAD) scanning between 400 and 750 nm. Pigments were identified by comparing characteristic absorbance spectra with that stored in a library created from a suite of standards obtained from DHI Denmark.

## 2.8 Statistics

To compare the similarity of different proxy records and reduce their overall dimensionality, we performed Principal Coordinate Analysis (PCoA) using a Bray-Curtis dissimilarity matrix on both the bulk geochemistry and algal pigment proxy datasets. All analyses were performed in the open-source platform R (R Core Team, 2021) using the phyloseq package (McMurdie and Holmes, 2013).

## 3 Results

### 3.1 Tephra stratigraphy and age model

Torfdalsvatn's tephra stratigraphy is shown in Figure 2 and Table 2, where age and origin are indicated along with correlations between the 2012NC/BC and 2004NC cores. Detailed tephra layer descriptions ($n = 33$), compositional datasets (>2200 grains



analyzed), and elemental plots used to identify the tephra layers are provided in the Supporting Data and Information. We stress, unless otherwise noted, that our chemical data is obtained from pristine basaltic to silicic tephra grains that feature delicate protrusions (e.g., Harning et al., 2024). In samples regarded as representing pristine tephra fall, the grains in the mafic realm (basalt to andesite) range from (i) non- to poorly (<20 %) vesicular, black blocky glass grains defined by brittle fracture surfaces to (ii) poorly- to highly-vesicular (20 to 60 %) black to brown glass grains with vesicle-pitted surfaces to (iii) highly vesicular (>60 %) translucent (sideromelane), brown to pale brown glass grains with very convolute outlines to (iv) achneliths defined by coherent shiny black glazed/fused outer surfaces (e.g., Walker and Croasdale, 1972; Thordarson et al., 1996). The felsic (high-silica andesite, dacite to rhyolite) realm are highly vesicular (>60 %) grains and are either pumice fragments with irregular outlines or fragments dominated by tube-vesicles or vesicle-wall/bubble-junction glass shards (e.g., Fisher and Schmincke, 1984).

The age model for core 2012NC uses 2 historic tephra layers of well-established age (Hekla 1766 and Hekla 1300, Thórarinsson, 1967), 14 PSV tiepoints, and 4 $^{14}$C-dated macrofossils (Fig. 3c). With the lowest PSV tie point assigned an age of 10868 cal a BP, the high density of chronological control points ($n$ = 20) results in an average of about 1 every 500 years. Combined with the Bayesian modeling approach, our age model therefore provides relatively precise and high-quality estimates on the ages of new tephra layers as well as the timing of past paleoenvironmental change derived from proxy records (e.g., Blaauw et al., 2018). Based on this age model, sedimentation accumulation rates (SAR) have minimal variation throughout the entirety of the Holocene record (Fig. 3d). Due to the near linear sedimentation rate, we do not present the following proxy data as fluxes as this results in negligible changes to the structure of proxy records.

**Table 2: Summary of Torfdalsvatn tephra stratigraphy and chronology. See Supporting Data for major element compositions and chemical identification of the tephra layers.**

| Tephra ID[1] | Dominant composition | Cumulative depth (cm)[2] | Thickness (cm) | Age (cal a BP) | Number of events[4] |
|---|---|---|---|---|---|
| H 1766* | Rhyolite to icelandite | 15 | 0.1 | 184 | 3 |
| H 1300* | Rhyolite to icelandite | 66 | 0.1 | 650 | 3 |
| **H 1104*** | Rhyolite | 84 | 0.7 | 846 | 3 |
| C 990 | Mix | 104 | - | 990 ± 80 | 4 |
| C 1180 | Mix | 125 | - | 1180 ± 70 | 2 |
| K 1220 | Alkali basalt | 131 | 1 | 1220 ± 50 | 3 |
| K 1270 | Alkali basalt | 142.5 | 2 | 1270 ± 40 | 2 |
| C 1850 | Mix | 202 | - | 1850 ± 50 | 3 |
| K/G 1990 | Mix | 219 | 0.4 | 1990 ± 140 | 3 |
| C 2320 | Mix | 257 | - | 2320 ± 70 | 4 |
| H C* | Icelandite | ~340 | 0.5 | 2800 ± 80 | 2 |
| **H 3*** | Icelandite, rhyolite | ~349 | 2 | 3060 ± 30 | 2 |
| H 4* | Rhyolite | 391 | 1.3 | 4260 ± 10 | 2 |
| H 4270 | Alkali basalt | 397 | - | 4270 ± 180 | 2 |
| **H 5100** | Rhyolite and alkali basalt | ~450 | 0.8 | ~5100 | 1 |
| **A 5700** | Primitive basalt | ~505 | 0.1 | ~5700 | 1 |
| Kv/K 5850 | Tholeiite basalt | 512 | 0.1 | 5850 ± 200 | 2 |
| **K 6500 (Tv-5)** | Alkali basalt | 557 | 1.1 | 6490 ± 130 | 1 |



| G/K 8500 | Tholeiite/alkali basalt | 675 | 0.1-0.2 | ~8500 | 2 |
|---|---|---|---|---|---|
| G? 9260 | Tholeiite basalt | 730 | 0.3 | 9260 ± 300 | 1 |
| G 9410 (G10ka Series #13)* | Tholeiite basalt | 734 | 0.6 | 9410 ± 340 | 2 |
| G 9630 (G10ka Series #12)* | Tholeiite basalt | 739.5 | 1.1 | 9630 ± 350 | 1 |
| G 9740 (G10ka Series #11)* | Tholeiite basalt | 742.5 | 0.5 | 9740 ± 320 | 2 |
| G 9850 (G10ka Series #10)* | Tholeiite basalt | 746 | 0.9 | 9850 ± 300 | 2 |
| G 9960 (G10ka Series #9)* | Tholeiite basalt | 749 | 1 | 9960 ± 240 | 1 |
| G-10120-10400 G10ka Series (1-8) (Tv-4)* | Tholeiite basalt | 756-781 | 25 | 10120-10400 | 7 |
| H 10550 | Alkali basalt | 802 | 0.3 | 10550 ± 150 | 2 |
| I-THOL-I? (Tv-3)* | Tholeiite basalt | 804 | 0.3 | 10560 ± 150 | 2 |
| Kv 10630 | Tholeiite basalt | 812.5 | 0.3 | 10630 ± 150 | 3 |
| K 11170 | Rhyolite, intermediate, alkali basalt | 827 | 0.4 | 11200 ± 330[3] | 3 |
| K 11295 | Rhyolite, intermediate, alkali basalt | 836 | 0.9 | 11360 ± 340[3] | 1 |
| K 11315 (Tv-2) | Rhyolite, intermediate, alkali basalt | 837 | 1.1 | 11375 ± 330[3] | 1 |
| H 11390 (Tv-1) | Alkali basalt | 845 | 0.6 | 11390 ± 180[3] | 1 |

[1] Abbreviation in the tephra ID column are as follows: A, Askja: C, cryptotephra; G, Grímsvötn; H, Hekla; K, Katla; Kv, Kverkfjöll; V-B, Veiðivötn-Bárðarbunga.
[2] Refers to depth in 2012NC sediment core.
[3] Ages reported in Harning et al. (2024).
[4] Indicates number of tephra fall events represented within individual tephra horizons. Total is 78.
*Established marker tephra layers from Iceland.
Bold ID indicates that residual sulfur content was measured in the groundmass glass of the tephra.
Underlined ID indicates potential regional markers.

## 3.2 Bulk geochemistry

Due to the high sampling density of bulk geochemistry ($n = 407$) and BSi ($n = 284$), the average temporal resolution of Torfdalsvatn's record equates to 1 sample every 27 and 39 years, respectively. Torfdalsvatn's magnetic susceptibility (MS) records shows the highest values at the base of the sediment core, which subsequently decrease to relatively low and stable values around 7000 cal a BP before rising again at 1350 cal a BP (Fig. 4a). %TOC ranges from 0.85 to 10.44 % with lowest values at the beginning of the record and highest values in the most recent portion (Fig. 4b). C/N values range from 5.33 to 11.13 with relatively high values from the beginning of the record to 7400 cal a BP, followed be a minimum between 7300 and 1080 cal a BP, before rising to reach a second maxima during the last several centuries (Fig. 4c). $\delta^{13}$C values range from -23.05 to -15.87 ‰ and show large centennial-scale variability with the most $^{13}$C-depleted values occurring near the base and top of the record and broadly anticorrelated with C/N (Fig. 4d). Along with C/N values, $\delta^{13}$C values are consistent with organic matter originating from a combination of aquatic and terrestrial sources, where their coincident changes (C/N increase and $^{13}$C-depletion) at 1800 cal a BP are consistent with a shift to a larger proportion of terrestrial organic matter to the lake (e.g., Geirsdóttir et al., 2020) (Fig. 4c-d). Finally, BSi ranges from 77 to 150 absorbance units, with the lowest values at the base and top of the record, and relatively higher but variable values in between (Fig. 4e).





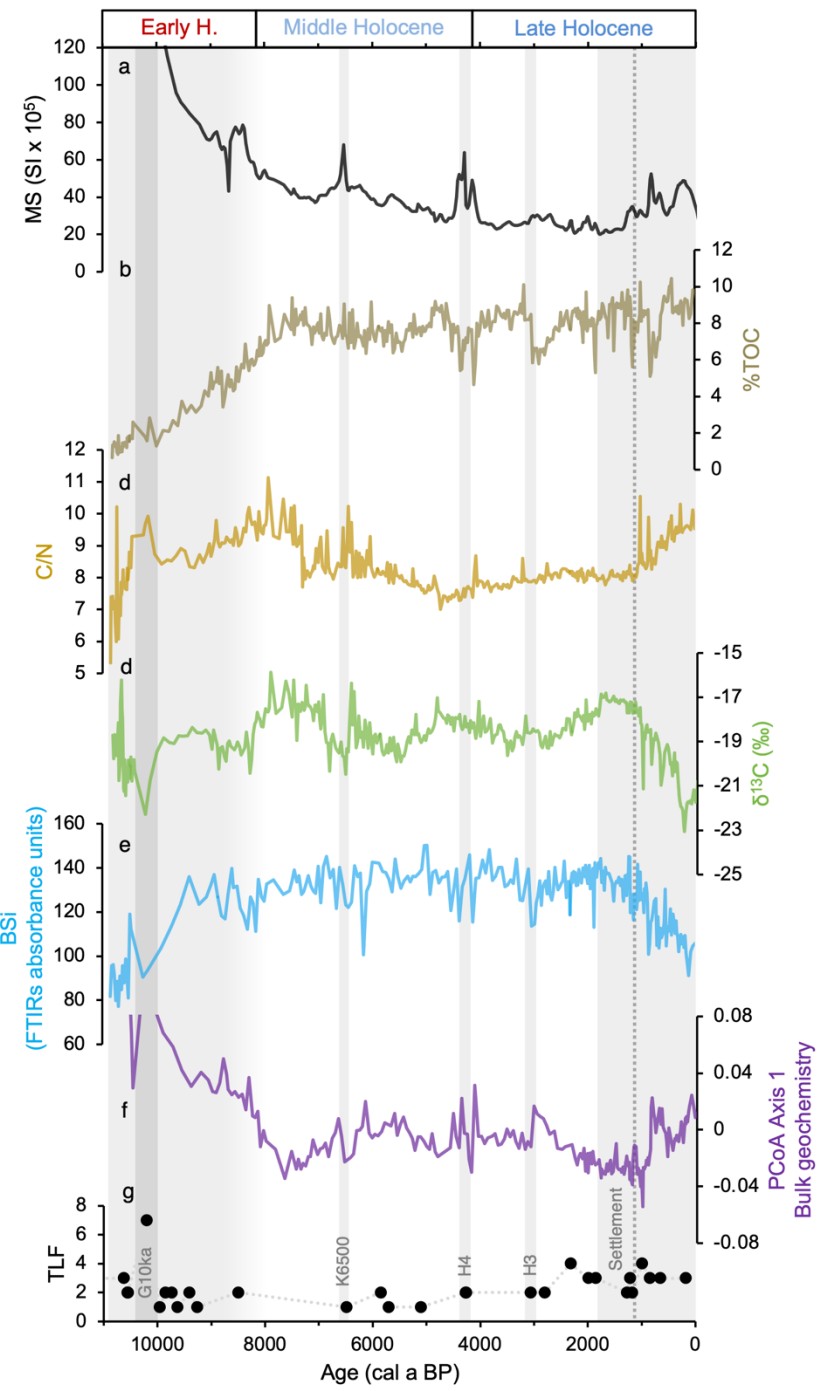

230    **Figure 4: Bulk physical and geochemical proxy records (a-f), PCoA Axis 1 of bulk geochemistry proxies (G), and tephra layer frequency (TLF, H) from core 2012NC. Note that MS in panel a is truncated at 120 SI x 10⁵ and values at the base of the record reach up to 523 SI x 10⁵. Grey bars reflect landscape disturbances associated with large tephra layer deposits and erosion and dashed gray line denotes the timing of presumed human settlement (1080 cal a BP).**



### 3.3 Algal pigments

The average temporal resolution of Torfdalsvatn's algal pigment sampling ($n = 246$) equates to 1 sample every 44 years. Algal pigment results are presented as the concentration of total chlorin (chlorophyll + degradation products), concentration of diatom pigments (fucoxanthin + diatoxanthin + diadinoxanthin), lutein to diatoxanthin ratio (L/D, ratio of green algae and higher plants to diatoms and chrysophytes, Leavitt and Hodgson, 2001), and the concentration of the cyanobacterial pigment canthaxanthin (Fig. 5). In all samples measured, chlorins are the most abundant pigment (Fig. 5a), making up between ~65 and 95 % of total pigments detected. In the earliest part of the record, chlorin concentrations are at their highest and rapidly decreasing until 9200 cal a BP with a subsequent period of relatively elevated concentrations between 6030 and 3020 cal a BP (Fig. 5a). The proportion of diatom pigments is variable, ranging from ~1 to 27% of total pigment with the lowest relative abundance of diatom pigments occurring at the beginning of the record (Fig. 5b). After peaking at 8720 cal a BP, diatom pigment abundance begins to decrease, and along with anticorrelations with L/D, suggests that green algae became more abundant (Fig. 5b-c). Lutein-producing (green) species remain dominant, and the relative proportion of diatom pigments is low until 5560 cal a BP (Fig. 5b-c). Between 5560 and 1800 cal a BP, diatom pigments steadily increase, and L/D stays low (Fig. 5b-c), suggesting elevated diatom relative abundance through this interval before a change in sign in both proxies (low diatom pigments and increased L/D) during the last 1800 years. The only detectable pigment biomarker of cyanobacteria in Torfdalsvatn is canthaxanthin, present throughout the core at low relative abundance between ~0.5 and 2.5% of total pigment (Fig. 5d). *Nostoc* sp. were commonly observed on shoreline rocks while sampling the epilithon in July 2014, and may therefore be a likely source of canthaxanthin. Except for the highest concentrations at the base of the record, cyanobacterial pigments are relatively low until 8600 cal a BP, then steadily increase until 8040 cal a BP (Fig. 5d). Canthaxanthin concentrations remain high until ~5450 cal a BP, albeit with some variability, decreasing thereafter until reaching minimum values during the last 500 years (Fig. 5d). Post-depositional diagenesis does not appear to control any of the observed trends as pigment concentrations do not systematically decrease downcore and the highest concentrations are found in the oldest sediments.



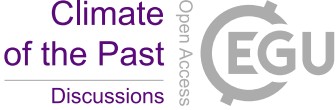



**Figure 5: Algal pigment proxy records (a-d), bulk geochemical BSi (e), PCoA Axis 1 of algal pigment proxies (f), and tephra layer frequency (TLF, g) from core 2012NC. Grey bars reflect landscape disturbances associated with large tephra layer deposits and erosion and dashed gray line denotes the timing of presumed human settlement (1080 cal a BP).**

### 3.4 Statistics

PCoA results for the bulk geochemistry and algal pigment proxy datasets show that axis 1 explains 52.4 % and 71 % of the variance, respectively. PCoA Axis 1 for bulk geochemistry proxies shows the highest values at the base of the record followed be a gradual decrease to a low at 7650 cal a BP (Fig. 4f). During the Middle Holocene, the PCoA values show some centennial scale variability, although millennial scale variability is relatively stable before a step shift at 1080 cal a BP (Fig. 4f). The major changes observed, such as the Early Holocene decrease, Middle Holocene centennial-scale variability and step shift at 1080 cal a BP, coincide with either erosional signals and/or tephra deposits (gray bars, Fig. 4). PCoA Axis 1 for algal pigments closely resembles total diatom and chlorin pigments and shows relatively large centennial scale variability compared to the relatively stable millennial scale variability. Major deviations in the trends include a low from ~7250 cal a BP to 4950 cal a BP and a steady decline from 1800 cal a BP towards present (Fig. 5f). We interpret PCoA Axis 1 of bulk geochemistry and algal pigments to reflect relative changes in erosional activity and total algal productivity, respectively.

### 4 Discussion

### 4.1 Holocene tephra records

During the Holocene, Icelandic volcanism has featured both effusive and explosive mafic eruptions, where the latter have often been influenced by frequent subglacial and subaqueous/submarine interactions. Although less common, rhyolitic eruptions from central volcanoes are frequent in Iceland and typically more explosive than their mafic counterparts (e.g., Thordarson and Larsen, 2007; Thordarson and Höskuldsson, 2008; Larsen and Eiríksson, 2008). Due to their added intensity, many of these rhyolitic eruptions have produced widespread, light-colored tephra layers that form the backbone of tephrochronological frameworks on both sides of the North Atlantic (e.g., Abbott and Davies, 2012; Lawson et al., 2012). Most of these tephra layers have been dated in soil and sedimentary sections via radiocarbon ($^{14}$C) and soil/sediment accumulation rates (Mangerud et al., 1984, 1986; van den Bogaard et al., 1994, 2002; Dugmore et al., 1995; Pilcher et al., 1995, 1996; Birks et al., 1996; Wastegård et al., 2001; Langdon and Barber, 2001; Bergman et al., 2004; Óladóttir et al., 2005, 2007, 2011; Gudmundsdóttir et al., 2011, 2016, 2018; Timms et al., 2017, 2018; Harning et al., 2018b, 2019) in addition to annual layer counting in Greenland ice cores (Grönvold et al., 1995; Rasmussen et al., 2006). However, despite the vast existing literature and the high frequency of tephra-producing eruptions from Iceland during the Holocene (n ≈ 2400; Thordarson and Höskuldsson, 2008), the Icelandic record is incomplete due a combination of preservation and the relatively few records with complete tephra layer inventories (e.g., Larsen et al., 2002; Thordarson and Larsen, 2007). Given Iceland's proximity to major regions of paleoenvironmental research, and the community's growing reliance on its tephra layers for age control and correlation



290 between sites, continuing to work towards a well-dated and compositionally well-defined master tephra stratigraphy is paramount.

We identified 28 visible tephra and 5 cryptotephra horizons in the lake sediment of Torfdalsvatn. In total, these tephra horizons represent at least 78 separate fall events (Table 2) as the majority (*n*=24; 73%) feature tephra fall from more than one volcanic system. On 9 occasions (27%), the horizons contain tephra with composition indicating a single system as the source.

295 Most common are tephra horizons with 2 source systems (*n*=13; 39%), then those with 3 source systems (*n*=8; 24%), followed by 4 (*n*=2; 6%) and then 7 (*n*=1; 3%). In 19 out of 24 (79%) tephra horizons, the tephra populations indicate additional source systems present as minor components, and in some instances only by a single grain. In 6 instances, the different source system populations are present in more equal proportions and 4 of those cases are represented by the cryptotephra horizons. The source systems indicated are Grímsvötn (*n*=31; 40%), Hekla (*n*=17; 22%), Katla (*n*=14; 19%), Veiðivötn-Bárðarbunga (*n*=7; 9%),

300 Askja (*n*=3; 4%), Kverkfjöll (*n*=3; 4%), and unknown (*n*=2; 3%) (SDT2). All these systems are distal to Torfdalsvatn, where Katla is farthest away (~270 km) and Askja is closest (~200 km). Consequently, only the more powerful explosive eruptions at these systems had the potential to produce tephra fall at Torfdalsvatn. Please see Supporting Information for more details on the implications of Torfdalsvatn's tephra stratigraphy.

Most lake sediment records in Iceland that report tephra compositional analyses only do so for key marker tephra to 305 provide age control for accompanying proxy records. Currently, there are 6 regional lake records that provide complete tephra stratigraphies and chronologies for all tephra identified in the sedimentary sequences (Fig. 6). Compared to these other complete tephra stratigraphies, the 78 volcanic events identified in Torfdalsvatn are relatively high and only superseded by the lake Lögurinn record, which contains 149 volcanic events based on compositional analyses (Fig. 6, Gudmundsdóttir et al., 2016). Surprisingly, the tephra record from Hestvatn, which is located close to many of the active volcanic centers in south 310 Iceland only archives 38 distinct volcanic events (Fig. 6, Geirsdóttir et al., 2022). The records from west (Haukadalsvatn) and northwest Iceland (Vestfirðir) contain 37 and 27 tephra layers, respectively (Harning et al., 2018, 2019). While various factors can influence the preservation of tephra layers in individual lake sediment records (e.g., Boygle, 1999), the simplest explanation for the regional variability in tephra layer occurrence is the direction of prevailing atmospheric winds around Iceland. In the stratosphere (7-15 km altitude), where explosive ash plumes are predominately injected to, the prevailing winds 315 are westerly (Lacasse, 2001). However, if ash plumes are in the troposphere (<7 km altitude), upper stratosphere (>15 km altitude), or occur during the spring/summer when the prevailing stratospheric westerlies shift to weak easterlies (Lacasse, 2001), tephra can be carried westward.

In addition to the established Holocene tephra marker layers in the Icelandic record (highlighted with an * in Table 1), our high-resolution age model allows 14 additional tephra layers in Torfdalsvatn to serve as regional marker horizons in 320 north Iceland. These include 1) the thick and closely-spaced Late Holocene basaltic Katla 1220 and Katla 1270 tephra layers, 2) the Middle Holocene sequence of Hekla 5100, Kverkfjöll/Katla 5850, Askja 6100 and Katla 6500 (Tv-5) series, 3) the Early Holocene Grímsvötn/Katla tephra pair G/K 8500 and Grímsvötn 9260, and 4) the pre-G10ka Series basaltic Hekla tephra



layers, Hekla 10,550 and Hekla 11,390 (Tv-1), and the bimodal Katla tephra layers (Katla 11,170, Katla 11,295 and Katla 11,315, see Harning et al., 2024).

Finally, in this study, the residual sulfur content was measured in a selected suite of tephra samples (bold font, Table 1) to assess whether the events that produced those tephra layers involved interaction with external water upon eruption. Evidence of such interaction is an indicator of wet vent environment and thus a proxy for eruptions from within glaciers or through standing body of water (lake or the sea). We refer the reader to the Supporting Information for more details.

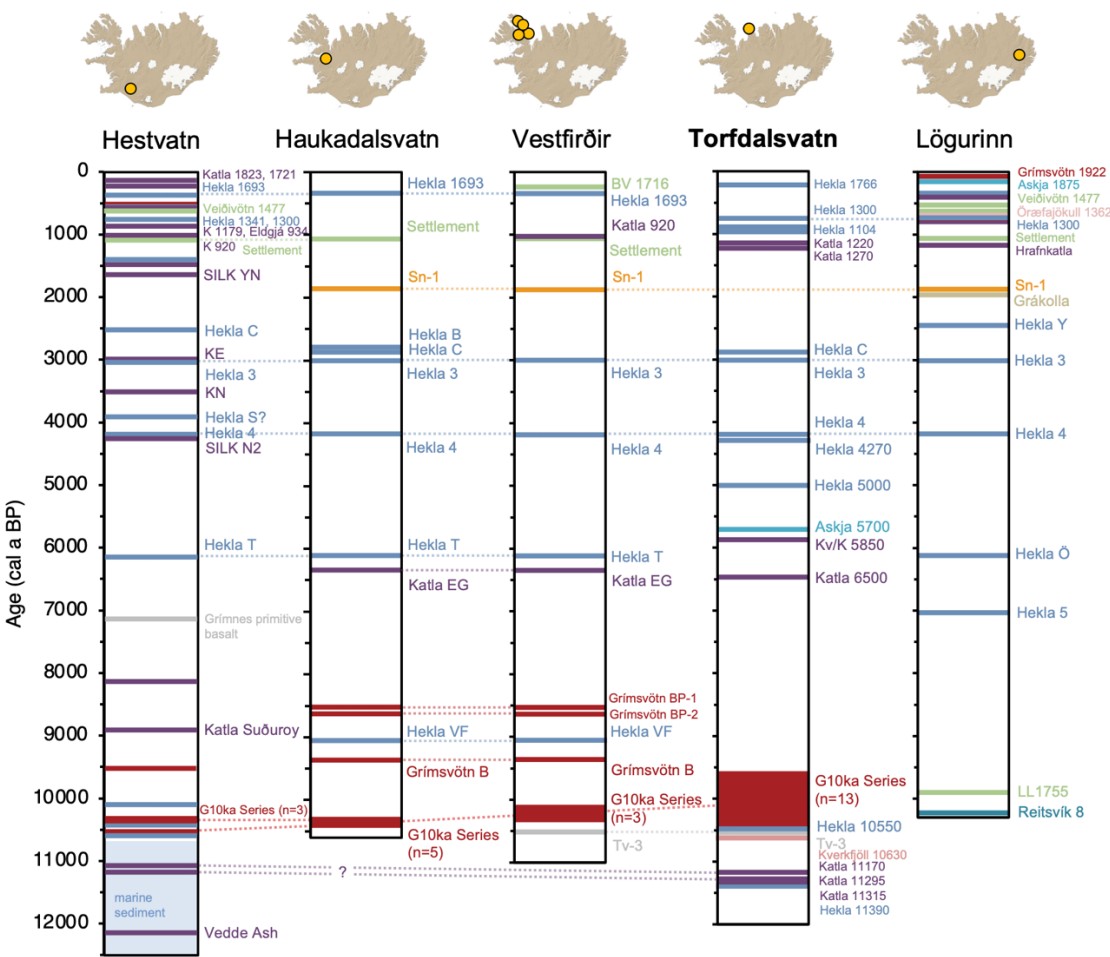

**Figure 6: Regional Icelandic tephra stratigraphies based on high-resolution lake sediment records from Hestvatn (Geirsdóttir et al., 2022), Haukadalsvatn (Harning et al., 2019), Vestfirðir (Harning et al., 2018), Torfdalsvatn (this study, Björck et al., 1992; Alsos et al., 2021; Harning et al., 2024), and Lögurinn (Gudmundsdóttir et al., 2016). For lake Lögurinn (east Iceland), we note that 149 total tephra layers have been identified and compositionally analyzed (Gudmundsdóttir et al., 2016), although we only show the key marker tephra layers for simplicity. All tephra layers are colored according to volcanic systems shown in Fig. 1a, and correlations between records indicated by dashed gray lines. Note: ages for tephra layers younger than the Settlement layer at ~1080 cal a BP (i.e., historical tephra) are presented in CE and not BP.**



## 4.2 Holocene climate and landscape evolution

Continuous records of past landscape stability and soil erosion can be reconstructed by comparing proxies for minerogenic

flux (magnetic susceptibility, MS) and terrestrial versus aquatic organic matter source (C/N) (e.g., Geirsdóttir et al., 2009, 2013, 2019, 2020). The controlling factors on the quantity of magnetic material that enters the lake are the amount of fine-grained minerogenic material available (from receding glaciers or tephra deposition), how well stabilized this material is by vegetation, and the strength of a transport mechanism (erosion by wind or water). In Iceland, organic matter source can be distinguished as either terrestrial or aquatic via C/N and $\delta^{13}$C composition (Wang and Wooller, 2006; Skrzypek et al., 2008;

Langdon et al., 2010; Florian, 2016), where terrestrial plants typically have high C/N (10 to 205) and low $\delta^{13}$C values (-31 to -22 ‰) and aquatic plants have low C/N (7 to 26) and high $\delta^{13}$C values (-30 to -11 ‰). Bulk sediment values therefore reflect the relative biomass in each environment, as well as efficiency of the transport of stored terrestrial material by soil erosion, which in Iceland, is easily accomplished by both wind and water due to the lack of cohesion of andosol soil (Arnalds, 2015). An increase in magnetic material (here inferred from magnetic susceptibility) without associated increase in C/N represents

either an increase in available source material (such as tephra) or a relatively small carbon-producing biomass that reduces dilution of minerogenic material.

Biogenic silica (BSi), which reflects the relative amount of siliceous material produced by lake algae (mostly diatoms), is a commonly used measure for reconstructing total aquatic productivity (Conley and Schelske, 2001), and in Iceland, spring temperatures as well (Geirsdóttir et al., 2009). On the other hand, algal pigments provide a more comprehensive

view of past aquatic productivity as they reflect all taxonomic groups, rather than just those that produce silicified structures (Leavitt and Hodgson, 2001), and primarily respond to broad changes in climate and nutrients (Smith, 1979; Smol and Cumming, 2000). Algal pigment concentrations and ratios, such as those present here, can be used to characterize both total aquatic productivity and the relative contribution of major algal groups as most pigments used are relatively stable, reducing the bias of variable pigment preservation through time (Bianchi et al., 1993; Leavitt and Hodgson, 2001).

### 4.2.1 Deglaciation to Early Holocene (~12000 to 8200 cal a BP)

Shortly after deglaciation, the receding Icelandic Ice Sheet left abundant, easily erodible minerogenic material upon the landscape. Plant sedaDNA records from Torfdalsvatn indicate that the first pioneer forb, graminoid and bryophyte taxa were present in the catchment no earlier than 12,000 cal a BP (Fig. 7c, Alsos et al., 2021; Harning et al., 2024). The lack of a substantial terrestrial ecosystem needed to stabilize the landscape at this time is consistent with the highest MS values in

Torfdalsvatn's Holocene record (Fig. 4a). MS then decreases as the quantity of easily erodible material was depleted and vegetation cover developed (Alsos et al., 2021). Woody taxa, such as *Salix* and *Betula*, then appear in the lake catchment by 10,300 and 9500 cal a BP, respectively (Alsos et al., 2021; Harning et al., 2023), straddling the deposition of the G10ka Series tephra and during the dominant period of terrestrial plant colonization (Alsos et al., 2021). MS values continue to steadily decrease through the G10ka Series tephra, although C/N increases, indicating a period of increased plant and soil erosion





around Torfdalsvatn that has been previously associated with the massive amount of tephra fallout from the Grímsvötn eruptions (Rundgren, 1998; Hallsdóttir and Caseldine, 2005; Eddudóttir et al., 2015; Florian, 2016).

Throughout the Holocene record, chlorins are the most abundant algal pigment in Torfdalsvatn, consistent with their ubiquity in photosynthetic organisms and function as light-harvesting pigments (Simkin et al., 2022). In the Early Holocene, chlorin concentrations peak at 10760 cal a BP (Fig. 5a). At this time, significant terrestrial biomass had yet to develop on the
landscape (Alsos et al., 2021), meaning most organic matter was aquatic and algal pigment-rich, which is supported by low C/N and high $\delta^{13}C$ values in bulk sediment (Fig. 4c-d). This is also consistent with a high proportion of unidentified *sed*aDNA reads, which is presumed to derive from algae due to limited reference material currently available (Alsos et al., 2021). The subsequent decrease in chlorin concentrations (Fig. 5a) therefore likely reflects terrestrial biomass development (Alsos et al., 2021) and influx of algal pigment-poor material. This interpretation is supported by increasing C/N ratios, which indicate
increased proportion of terrestrial carbon through this interval (Fig. 4c). The proportion of diagnostic pigments from algal groups (e.g., diatoms, L/D, and canthaxanthin) are variable during the Early Holocene. Diatom pigments become more abundant after the deposition of the G10ka Series tephra and an increase in BSi during this interval suggests an increase in diatom productivity (Fig. 5b and e). These changes occur during a period of broad summer warming in Iceland, most clearly manifested in the rapid retreat and likely disappearance of residual Icelandic ice caps (Larsen et al., 2012; Anderson et al.,
2019), including Drangajökull (Harning et al., 2016b), located ~75 km to the west of Torfdalsvatn (Fig. 1a). After peaking at 8720 cal a BP, diatom pigment concentrations decrease, anticorrelated with the ratio of lutein to diatoxanthin (Fig. 5b-c). This, along with a decrease in C/N (Fig. 4c), suggests that green algae and/or aquatic higher plants became increasingly more abundant at the expense of diatoms.

During the Early Holocene, marine sediment proxy records from the North Iceland Shelf indicate that surface currents
were generally dominated by warm Atlantic Water and restricted sea ice (e.g., Kristjánsdóttir et al., 2017; Xiao et al., 2017; Harning et al., 2021) (Fig. 7). However, additional lake sediment and mire records from north Iceland document an interruption in woody taxa succession and increased soil erosion from ~8800 to 7900 cal a BP (e.g., Hallsdóttir, 1995; Hallsdóttir and Caseldine, 2005; Eddudóttir et al., 2015, 2018; Geirsdóttir et al., 2020). This time window includes the well-known 8.2 ka event (e.g., Barber et al., 1999; Alley and Ágústsdóttir, 2005; Rohling and Pälike, 2005) as well as additional freshwater pulses
that originated from the decaying Laurentide Ice Sheet (Jennings et al., 2015) that are known to have driven oceanographic cooling around Iceland via a slowdown of the North Atlantic ocean circulation (e.g., Quillmann et al., 2012; Moossen et al., 2015). PCoA results for Torfdalsvatn soil erosion demonstrate that while the catchment was in transitional state towards stabilization (Fig. 7a), algal productivity generally decreases in response to cooling in the ~8800 to 7900 cal a BP window (Fig. 7b). In terms of vegetation, plant *sed*aDNA species richness show little disturbances to some plant functional groups at
this time, whereas others, such as graminoids and aquatic plants may have decreased. However, we note that this lack of response in some plant groups may partially be due to the low resolution of the record (>250 years, Fig. 7c).





**Figure 7: Comparison of Torfdalsvatn's PCoA results with regional marine and terrestrial environmental records. a) Torfdalsvatn PCoA Axis 1 of bulk geochemistry (this study), b) Torfdalsvatn PCoA Axis 1 of algal pigments (this study), c) Torfdalsvatn species**
**richness of various plant functional groups based on *sed*aDNA (Alsos et al., 2021), d) relative abundance of Arctic planktic foraminifera *N. pachyderma* from marine core MD99-2269 (Harning et al., 2021), e) concentration of sea ice algae biomarker IP$_{25}$**



from marine core MD99-2269 (dark blue, Cabedo-Sanz et al., 2016) and MD99-2272 (light blue, Xiao et al., 2017), and f) Drangajökull ice cap advances (blue, Harning et al., 2016a, 2018a) and mean summer air temperature anomaly (ºC) from Skorarvatn, NW Iceland (teal) with uncertainty estimates (light teal) (Harning et al., 2020). Grey bars reflect landscape disturbances associated with large tephra layer deposits and erosion and dashed gray line denotes the timing of presumed human settlement (1080 cal a BP).

### 4.2.2 Middle Holocene (8200 to 4200 cal a BP)

Overall, MS continues to decrease through the Middle Holocene, except for periodic spikes at 6500 and 4260 cal a BP reflecting the presence and impact of major tephra fall from Katla and Hekla eruptions, respectively (Fig. 4a). At least for the Katla 6500 tephra, increased C/N and $\delta^{13}$C values indicate increased soil erosion immediately following its deposition, which lasted for several centuries (Fig. 4c-d). Hekla 4 has similarly disturbed the terrestrial landscape in other Icelandic lake catchments (Geirsdóttir et al., 2019), however, Eddudottir et al. (2017) suggest that lowland areas with more substantial *Betula* woodland cover are more resilient to tephra fall compared to higher elevation areas that were already at the climatic/ecological limit. The persistence of woody taxa around Torfdalsvatn through these large tephra fall events (Alsos et al., 2021) is consistent with this inference. However, we note that sampling resolution for *sed*aDNA in this portion of the record is over 100 years (Alsos et al., 2021), which may miss short-term impacts of tephra to the catchment ecosystem. In terms of aquatic proxies, we note that all algal pigments as well as BSi indicate short-lived increases in algal productivity during both the Katla 6500 and Hekla 4 eruptions. This process is consistent with lake studies in other volcanic regions that document increased diatom and algal productivity following volcanic eruptions due to the increased supply of silica and nutrients (e.g., Telford et al., 2004; Egan et al., 2019).

Chlorin concentrations remain generally low during first two millennia of the Middle Holocene before increasing to peak values at 4580 cal a BP (Fig. 5a). These higher chlorin values are associated with increasing diatom pigments (Fig. 5b), decreasing L/D ratios (Fig. 5c), and the lowest C/N of the record (Fig. 4c), indicating a shift of organic matter towards a more aquatic plant source around this time. Interestingly, BSi does not track diatom pigment abundance for much of the record. One possibility is that there are several complicating factors using both organic and inorganic indicators to reconstruct past algal biomass. Both these proxies (i.e., diatom pigments and BSi) can be influenced by differences in amount of each compound per unit of algal biomass, varying species assemblage, and environmental conditions (Alberte et al., 1981; Lavaud et al., 2002; Rousseau et al., 2002; Stramski et al., 2002; Finkel et al., 2010), all of which are challenging to individually constrain in paleoenvironmental reconstructions. In any case, during the Middle Holocene, terrestrial pollen (Rundgren, 1998; Hallsdóttir and Caseldine, 2005; Eddudóttir et al., 2015) and plant *sed*aDNA records (Alsos et al., 2021) indicate that catchment vegetation communities were well-developed, which likely stabilized the catchment and reduced the influx of terrestrial organic matter to the lake. In addition, aquatic plants make up the dominant proportion of *sed*aDNA reads at this time (Alsos et al., 2021), which likely results from the small, shallow nature of the lake basin that permits light penetration for submerged taxa (Fig 1b), consistent with the increased general productivity of the aquatic environment.





Finally, cyanobacterial populations have been shown to quickly increase in response to higher temperature and nutrient levels and may therefore be an important indicator species for past lake conditions (De Senerpont Domis et al., 2007; Paerl and Paul, 2012). The only detectable pigment of cyanobacteria in Torfdalsvatn is canthaxanthin, present throughout the core at low concentrations. Canthaxanthin concentrations remain elevated until ~5450 cal a BP, albeit with some variability,
decreasing thereafter until reaching minimum values during the last 500 years (Fig. 5d). The pattern of change in canthaxanthin concentrations mirrors other relative and quantitative temperature records derived from Icelandic lake sediment BSi composites and lipid biomarkers records, respectively, which peak during the Holocene Thermal Maximum (7900 to 5500 cal a BP) and decrease in a stepwise manner through the Middle Holocene to the Little Ice Age (700 to 50 cal a BP, 1250 to 1900 CE) (Larsen et al., 2012; Geirsdóttir et al., 2013, 2019; Harning et al., 2020). This suggests that the abundance of cyanobacteria
may be controlled more closely by lake water temperature and length of summer than the other algal groups in Torfdalsvatn, such as diatoms.

PCoA results from bulk geochemistry and plant *sed*aDNA species richness demonstrate that the Middle Holocene terrestrial landscape around Torfdalsvatn was generally stable, whereas algal productivity diminished between 7500 and 5000 cal a BP (Fig 7a-c). This terrestrial stability is consistent with the corresponding marine environment along the North Iceland
Shelf, where the ocean surface was dominated by warm Atlantic Water at this time (e.g., Kristjánsdóttir et al., 2017; Harning et al., 2021), reflected well by low abundances of Arctic planktic foraminifera (e.g., *N. pachyderma*, Harning et al., 2021) and sea ice algae proxies (e.g., IP$_{25}$, Cabedo-Sanz et al., 2016; Xiao et al., 2017) (Fig. 7d-e). The decrease in algal productivity between 7500 and 5000 cal a BP in the PCoA results largely stems from decreases in diatom pigments as well as minor decreases in chlorins at the expense of lutein-producing green algae and/or higher plants (Fig. 5). As there is no indication of
local climate change during the 7500 to 5000 cal a BP interval in either the Icelandic marine or terrestrial realms, it is logical that the changes were induced by internal lake processes, such as nutrient availability. However, without further proxy information, the processes behind the changes in algal productivity are currently challenging to diagnose.

### 4.2.3 Late Holocene (4200 cal a BP to present)

After about two millennia of relative stability in Torfdalsvatn, individual algal pigment concentrations, BSi, and algal pigment
PCoA results begin to decrease at 1800 cal a BP (Fig. 5). These consistent changes indicate broad decreases in total algal productivity likely driven by ambient changes in climate and/or shorter ice-free seasons that inhibit light availability. In contrast to other high resolution lake sediment C/N proxy records in Iceland (Geirsdóttir et al., 2019, 2020), we do not find an increase in soil erosion accompanying algal productivity changes around Torfdalsvatn. This is consistent with the relative stability of the various terrestrial plant functional groups in the lake catchment derived from *sed*aDNA (Fig. 7C, Alsos et al.,
2021). However, regime shifts in offshore marine climate records demonstrate that around this time the Polar Front advanced southward around Iceland, bathing the North Iceland Shelf with cool, sea-ice bearing Polar Water (Harning et al., 2021). This shift in ocean surface water source is reflected well by the increase in Arctic planktic foraminifera *N. pachyderma* (Harning et al., 2021) and sea ice algae proxy IP$_{25}$ (Cabedo-Sanz et al., 2016) from sediment cores on the North Iceland Shelf just north of



Torfdalsvatn (MD99-2269, Figs. 1A and 7d-e), and led to substantial summer cooling in Skorarvatn, NW Iceland, and the
marginal expansion of the Drangajökull ice cap, just west of Torfdalsvatn (Figs. 1a and 7f, Harning et al., 2016a, 2018a, 2020).

At 880 cal a BP, bulk geochemistry PCoA results indicates a substantial increase in landscape instability and soil
erosion around Torfdalsvatn (Fig. 7a). Norse settlers are estimated to have arrived in Iceland around 1080 cal a BP (~870 CE)
and *assumed* to have quickly impacted the landscape through woodland clearing and agricultural and pastoral farming that
prohibited natural plant regeneration (e.g., Thórarinsson, 1944; Arnalds, 1987; Hallsdóttir, 1987; Smith, 1995; Lawson et al.,
2007). However, these datasets are largely based on sedimentary records with coarser resolution and/or age control than
Torfdalsvatn. The plant *sed*aDNA record from Torfdalsvatn indicates that while some woody taxa, such as *Salix* and *Betula*,
are present through this interval, *J. communis* (juniper) disappears and species diversity of forbs decrease slightly earlier than
our bulk geochemistry record (1055 cal a BP, Fig. 7C, Alsos et al., 2021), which may partially be due to variable age control
between the two sedimentary records. We note, however, that while environmental impacts from settlers have been
independently confirmed in lake sediments on the Faroe and Lofoten Islands using fecal biomarkers and/or sheep *sed*aDNA
(D'Anjou et al., 2012; Curtin et al., 2021), these tools have so far proved challenging for use in Icelandic sediments and are
needed for confirmation (Ardenghi et al., 2024). In any case, it is unlikely that the small populations that initially settled
Iceland (~30,000 people, *Landnámabók*, i.e. The book of Settlement) would have had a substantial and immediate impact on
the landscape, and our record from Torfdalsvatn provides a well-constrained benchmark for changes in the ecosystem possibly
linked to human practices in north Iceland.

Following the initiation of increased long-term soil erosion that began around 1080 cal a BP, we observe distinct
disturbances to the Hekla 1104 CE tephra layer stratigraphy. In core 2004NC, Hekla 1104 CE tephra is present over a 15.4 cm
interval (88.1 to 103.5 cm depth) as three distinct and separate layers/wedges connected by mm-thick, irregular light-colored
stringers with intermittent shear planes (Fig. 8). The 30 cm above the uppermost wedge is characterized by fining upward
pumice grains suggesting an origin within a gravity current (Fig. 8). In core 2012NC, it is present as two distinct layers at
depths of 80.5 cm (0.1 cm) and 84.5 cm (0.7 cm) (see Supporting Information), where the upper layer has a modeled age of
1150 CE. Hekla 1104 CE is also found in a previously published record from Torfdalsvatn (Alsos et al., 2021), where the
stratigraphy is also discontinuous with stringers below and above the main tephra layer horizon (Bender, 2020). The cause of
this disturbance is unlikely to relate to sediment coring as it is found in the middle of the core segments between undisturbed
500   organic lake sediment and identified in three independent records. Instead, this indicates that an external factor likely impacted
the sedimentary sequence after the Hekla 1104 CE tephra layer was deposited, such as a slope failure. As the catchment and
bathymetry of Torfdalsvatn are both low relief (Fig. 1b, Florian, 2016), a slope failure would require a trigger event. Local
documentary records indicate that a large earthquake occurred in 1260 CE (Storm, 1977a, 1977b, 1977c), which we propose
as a likely candidate based on the slightly younger age compared to the tephra deposit. While the disturbance is relatively
505   short-lived, the Torfdalsvatn records indicate the susceptibility of even low-relief environments to seismic activity that may
confound the continuity and interpretation of lake sediment proxies in Iceland.



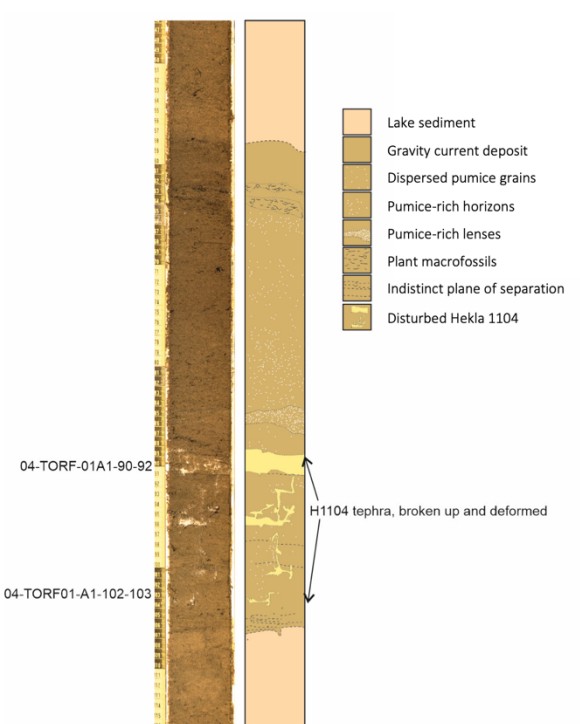

**Figure 8: Lithostratigraphy of the deformed Hekla 1104 CE tephra layer in core 2004NC. On the left is a photo image and on the right is simplified schematic based on the major identified sediment types.**

Finally, the general cooling reflected in decreased algal productivity that commenced at 1800 cal a BP culminated around 450 BP (Fig. 7b). This coincides with the Little Ice Age (LIA, ~700 to 100 cal a BP, blue bar, Fig. 7, Geirsdóttir et al., 2013), a period marked by the coolest Holocene conditions in Iceland. Regionally, the Little Ice Age is well reflected in environmental conditions such as the maximum Holocene extent of Polar Water and sea ice on the North Iceland Shelf (Fig. 7d-e, Cabedo-Sanz et al., 2016; Harning et al., 2021) and largest dimensions of Icelandic ice caps (Fig. 7f, Larsen et al., 2011; Harning et al., 2016b; Hannesdóttir et al., 2015, 2020). In terms of Torfdalsvatn plant communities, species richness patterns show troughs in forbs and aquatic taxa and peaks in graminoids and bryophytes during this interval (Fig. 7c). Alsos et al. (2021) also note an increase of high arctic taxa such as *O. digyna*, *K. islandica*, *Epilobium anagallidifolium*, *S. acaulis*, and *D. octopetala* at Torfdalsvatn during the Little Ice Age. Despite these broad changes in Torfdalsvatn's terrestrial and aquatic environments during the coldest part of the Holocene – that would be expected to result in enhanced landscape instability as observed in other Icelandic lake records (e.g., Geirsdóttir et al., 2009, 2020) – we observe a decrease in soil erosion between 650 and 100 cal a BP (Fig. 7a). The northern latitudinal position of the lake abutting the harsh Polar marine climate on the North Iceland Shelf and minima in Northern Hemisphere summer insolation (Berger and Loutre, 1991) would have resulted in shorter summers and possibly reduced mobilization of soil in a more perennially frozen landscape.



## 5 Conclusions

We present a multi-proxy analysis of lake sediments from Torfdalsvatn, the longest known Holocene terrestrial record in Iceland of ~12000 years. These analyses include a high-resolution age model ($n$ = 22 control points), an expanded and comprehensive Holocene tephra stratigraphy and chronology (>2200 grains analyzed in 33 tephra horizons), and sub-centennial bulk geochemical and algal pigment proxy records. Collectively, we use these datasets to address the following topics:

- *Tephrochronology:* We identified 33 distinct tephra layers, which represent 78 separate volcanic events from 6 volcanic systems. Compared to tephra stratigraphies from other lake sediment records in Iceland, these are relatively high numbers. Key marker tephra layers include I-THOL-I, the G10ka Series (13 events), Katla 6500, Hekla 4, Hekla 3, Hekla C, Hekla 1104, Hekla 1300, and Hekla 1766. In addition, we present evidence for a previously un-identified basalt tephra (Askja 6100) with a distinct primitive composition and a bimodal (rhyolite-basalt) tephra layer (Hekla 5100).

- *Catchment instability and soil erosion:* Bulk physical and geochemical proxy records capture past intervals of soil erosion around Torfdalsvatn due to combinations of climate cooling, Middle Holocene volcanic eruptions, and after human settlement at 880 cal a BP. Compared to other lake sediment records in Iceland, we do not observe long-term increases in soil erosion beginning prior to human settlement but starting ~200 years later. The stratigraphy of sediment during the last millennium also suggests the occurrence of a seismic-induced slope failure in or around the lake.

- *Algal productivity:* Torfdalsvatn's algal group ontogeny progressed from mainly diatoms shortly after the G10ka Series tephra, to predominantly green algae, aquatic macrophytes and cyanobacteria. This is assumed to have been driven by increased temperatures and length of summer during the HTM where the timing of peak cyanobacterial abundance likely represents the warmest Holocene temperatures in Torfdalsvatn. After the HTM, algal productivity remains generally stable until 1800 cal a BP, when changes in regional climate led to decreased algal productivity that reached the lowest values during the Little Ice Age (700 to 100 cal a BP).

- The changes in algal productivity and soil erosion observed beginning around 1800 and 880 cal a BP, respectively, highlight the impact of both natural and possibly anthropogenic factors on Late Holocene aquatic and terrestrial environmental changes in north Iceland. Importantly, they emphasize that while local climate was cooling prior to known human settlement, some low-elevation coastal regions such as Torfdalsvatn, may have been resistant to natural pre-settlement changes in vegetation cover and soil erosion than observed in other regions of Iceland.

### Data Availability

Data associated with this manuscript is available in the Supporting Information.



**Author Contributions**

ÁG and GHM conceptualized and funded the research; ÁG, GHM, CRF and YA acquired lake sediment cores; SÓ performed PSV analyses; DJH and TT performed EMP analyses and generated the age model; CRF analyzed bulk geochemistry and algal pigments; DJH, CRF, ÁG and TT wrote the paper with contribution from all co-authors.

**Competing Interests**

The authors declare that they have no conflict of interest.

**Acknowledgements**

We kindly thank Guðrún Eva Jóhannsdóttir for her contribution to electron microprobe measurements and Þorsteinn Jónsson, Sveinbjörn Steinþórsson, Jason Briner, Yiming Wang, and Matt Wooller for assistance in the field. DJH acknowledges support
from RANNÍS Doctoral Student Grant #163431051 and CRF acknowledges support from the Doctoral Grant of the University of Iceland. This project has been principally supported by the Icelandic Center for Research (RANNÍS) through Grant-of-Excellences #022160002-04, #70272011-13 and #141573051-3, the University of Iceland Research Fund, and the National Science Foundation ARCSS #1836981, awarded to ÁG and GHM.

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
