# Peer review of "High-resolution Holocene record based on detailed tephrochronology from Torfdalsvatn, north Iceland, reveals natural and anthropogenic impacts on terrestrial and aquatic environments"

_Climate of the Past, 2024_

## Community Comment (CC1)

[supplement omitted: unrelated document]

---

## Author Response (AR1)

**REVIEWER 1**

**A. Summary**

Harning et al. present a comprehensive tephro-stratigraphy and chronology (and tephra layer frequency), along with algal pigments (total chlorins, diatom pigments, and cyanobacterial pigment) and bulk geochemistry data (C/N, C-isotopes, bSi, TOC), from lake sediments cored in 2012 from the shallow Lake Torfdalsvatn in NW Iceland. This study documents one of the longest known continuous Holocene lacustrine archives in Iceland, with the record extending back to approximately 12 ka cal BP. Although Lake Torfdalsvatn has been the subject of many previous investigations, Harning et al. present new findings.

The manuscript is divided into two main sections: one focuses on the tephrochronology, including the identification of a previously unknown tephra, and the other examines the Holocene paleoenvironmental evolution of Torfdalsvatn and its watershed, including anthropogenic disturbances. In the first section, they present a thorough dataset containing 33 tephra layers representing approximately 78 volcanic events from six different source systems. They compare the supplemented Torfdalsvatn tephrostratigraphy with stratigraphies of four other Icelandic lakes.

In the second section, they analyze two datasets (bulk geochemistry and algal pigments) using PCoAs and interpret their first principal coordinates as proxies for soil erosion and algal productivity, respectively. These results are contextualized within local and regional climatic and environmental disturbances using previous reconstructions from Lake Torfdalsvatn and marine sediment cores near the northern margin of Iceland.

Despite the density of the manuscript, which presents extensive datasets that could constitute two independent papers, I found it engaging and insightful.

This study will be of interest to the readers of Climate of the Past.

I am pleased to recommend it for publication, pending the authors' addressing of some major (B.) and specific comments (C.).

We kindly thank Reviewer 1 for their time and consideration reviewing our manuscript that contribute to improving its clarity and focus. Below we address each individual comment in blue and identify where the associated edits have been made.

**B. Major Comments**

**Title**

- The title seems to overlook the tephra aspects of the manuscript, potentially hiding this significant dataset or misleading the reader. Consider revising the title to reflect the importance of the tephrochronology.

  Thank you for this suggestion. We have amended the title to highlight the tephra contribution of the paper: *High-resolution record based on detailed tephrochronology from Torfdalsvatn, north Iceland, reveals natural and anthropogenic impacts on terrestrial and aquatic environments.*

**Short summary**

- The short summary does not mention the tephrochronology. This important aspect needs to be included to accurately represent the manuscript.

We will amend the short summary to highlight the tephra contribution of the paper.

**Abstract**

- Here, you effectively highlight what appears to be the research gap: understanding the resilience of pre-settlement erosion. This could be one of the key topics to emphasize in the introduction.

We have edited the introduction accordingly. See L58-60.

**1 Introduction**

- The introduction provides a comprehensive review of significant literature; however, it lacks sufficient context for the tephrochronology, which is a substantial dataset presented in this paper. It would be beneficial to explain why revising the tephrochronology is important, the significance of adding new tephra layers, and the relevance of using lake sediments from NW Iceland. Similarly, addressing these questions and opening a clear research gap could highlight the importance of the other two datasets as well. Clearly outlining the research gaps could better frame the introduction and emphasize the study's important contributions, especially if it aims to become a regional template for North Iceland.

We have restructured the introduction to provide context for the tephrochronological work in the paper. See new paragraph L68-84.

**2 Methods**

- From an organizational perspective, I suggest merging chapters 2.3-2.5 into a section titled "2.3 Chronostratigraphic Techniques" or a similar name. More details below.

We have merged sections 2.3 to 2.5 together into a new Section 2.3 (Chronology). See L163-193.

**2.2 Tephra Sampling and Compositional Analysis**

- Ensure the inclusion of the Supporting Data Files. SDT1-3 were not available for the review, or the Discussion.

The supporting data files will be included in the revision, and if for some possible reason they cannot be there, certainly for the final publication.

**2.5 Chronology and Age Model**

- I suggest combining Sections 2.3-2.5 into a single section titled "2.3 Chronostratigraphic Techniques" or similar, with subsections as follows: "2.3.1 PSV," "2.3.2 Radiocarbon," and "2.3.3 Age Model." This approach will consolidate the short sections under the umbrella of "chronology" and reduce the overall number of sections in the methods.

We have merged sections 2.3 to 2.5 together into a new Section 2.3 (Chronology). See L163-193.

- Clarify whether the tephra layers, the tephra seismite (mentioned in a discussion comment), and other instantaneously deposited layers were removed and re-inserted for the age-depth model and SAR calculations. If this was done, explain the methodology in detail. If this was not done,

provide a rationale. Discuss the potential impact on tephra ages. Specify whether the top or bottom age was used to define the tephra layer ages (should be consistent if the layers were excised and re-inserted).

Most tephra layers were not removed from the age model due to their thin sub-cm thicknesses. Given the length of the sediment core of ~8 m, removing these small thicknesses will not affect the age estimates of new tephra or the proxy records within uncertainty of the age modeling code. Bottom ages were used for the tephra age estimate as this reflects when the tephra first fell on the landscape. See L191-193.

- Recap why it is reliable to include radiocarbon ages from aquatic macrofossils in the age-depth models in this region, considering factors like the absence of a hard-water effect and relevant geological and pedological features.

See L172-174.

**2.6 Bulk Geochemistry**

- How were the MS measurements of the different sediment core sections (for the 8m piston core) normalized to ensure comparability and to prevent artifacts when "glueing" the data back together? It is important to clarify this to ensure the PCoA results are not influenced by any inconsistencies.

The Geotek used to measure the MS used external calibration to ensure that individual sections of sediment core are comparable. See L207-208.

- Elaborate on the uncertainty associated with BSi measurements using FTIR. Indicate whether FTIR was calibrated against other BSi methods and specify the wavelengths used to extract semi-quantitative BSi values. Discuss potential interferences from other minerals that might explain the observed trend in the late Holocene.

These aspects have been clarified in the text. See L213-216.

**2.7 Algal Pigments**

- Were the 246 samples taken from the same depths as the bulk samples? Additionally, how much depth did they integrate per sample (e.g., 1 cm)?

- Specify the DAD-detectors model and the range (400-900nm?).
- Mention that pigment concentrations are reported in µg pigment/mg OC.

These points have been clarified in the revised manuscript. See L218-266.

**2.8 Statistics**

- Provide details on the pretreatment of data: how were the MS data sampled to match the resolution of the destructive samples?
- Specify the data included in PCoA1 and PCoA2. Describe the criteria used to determine the number of significant coordinates. Discuss how biplots (e.g., coordinate 1 vs. 2) reveal specific patterns and consider including this as a supplementary figure.

**3 Results**

- From an organizational perspective, I suggest removing section "3.4 Statistics" and integrating the PCoA1 results with section "3.2 Bulk Geochemistry" and the PCoA2 results with section "3.3 Algal Pigments". This approach aligns with the structure of Figures 4 and 5, respectively, and will provide better guidance for the reader.

We have restructured the results accordingly. See L329-335 and L388-392.

**3.1 Tephra Stratigraphy and Age Model**

- Line 200: While the SAR is generally low, note that the variations are significant as values can double. Thus, the term "minimal variation" is probably not appropriate.

While there are indeed "sharp changes" in the SAR, the range of SAR variability is relatively narrow (new panel b y-axis), particularly when compared to lake sediment sedimentation histories of other non-glaciated lakes in Iceland with SAR that can increase up to 1.5 cm/yr (e.g., Geirsdóttir et al., 2013). This aspect has been emphasized in the text. See L294.

- Line 202: If dry bulk density data are available (e.g., from the 407 samples) and MAR can be calculated, I suggest comparing concentrations and fluxes in a supplementary figure. MAR could provide additional insights into erosional patterns.

Due to the linear sediment rate, calculating fluxes does not alter the structure of the proxy curves and therefore does not provide any difference in plots than those shown in the figures presented. This point was already made in the original manuscript. See L300.

**3.2 Bulk Geochemistry**

- I suggest integrating the PCoA1 results presentation from the "3.4 Statistics" section here.

**3.3 Algal Pigments**

- I suggest integrating the PCoA2 results presentation from the "3.4 Statistics" section here.
- Line 237: Specify which degradation products were included, such as pheophytins, pheophorbides, pyro-pheophorbides, etc.
- Address the spike in Canthaxanthin around 9.8 ka BP is it considered an outlier?

**3.4 Statistics**

- As mentioned above, distribute the contents of this section to sections 3.2 and 3.3, respectively.
- Add a scatter (biplot) figure to the supplementary materials to better illustrate how the PCoAs explain the dissimilarity and the characteristics of PCoA axes 2.
- Line 263: Clarify whether the PCoAs explain the variance of the dataset or the dissimilarity. Probably reformulate to "explain XYZ% of the variance of the dissimilarity".

We have revised the section accordingly. See L329-335 and L388-392 and new supplemental figures S5 and S6.

**4 Discussion**

- From an organizational perspective, the section about tephra starting at L491 seems misplaced, as it is currently in the climate and landscape evolution chapter. Relocating the large fraction of this part to chapter 4.1 may streamline the discussion about human impact in section 4.2.3.

We appreciate this comment and have considered this suggestion during the revision, but ultimately decided to keep the paragraph in the current landscape evolution section. While the proxy is indeed a tephra layer, we feel that the inferred seismic activity from the layer is more appropriate here than in the tephra section that focuses on chronology.

**4.2 Holocene Climate and Landscape Evolution**

- Lines 345-346: Clarify the definition of low/high δ13C values. The current phrasing is confusing as "low" values (-31 to -22‰) fall within the range of "high" values (-30 to -11‰). Reformulate to indicate that values higher than -22‰ may predominantly indicate aquatic plants, while values lower than -22‰ may represent a mixture. Note that most of the organic matter throughout the Holocene appears to be aquatic, except for the topmost part of the record.

We have clarifiedthe section accordingly. See L570-571.

**4.2.2 Middle Holocene**

- Confirm whether the magnetic susceptibility (MS) of the tephra layers was included in PCoA 1. It is assumed that the MS of the tephra layers was not included, as the destructive samples were likely taken only from the background sediment. However, this is not explicitly stated. If MS was included in PCoA axis 1 and used as an erosion proxy, discuss whether the tephra layers influenced the interpretation of erosional patterns. Clarifying in sections 2.6 or 2.8 how you resampled the high-resolution continuous MS data to align with the discrete destructive samples, will tackle this issue.

We did not include MS in the PCoA analyses due to the interference of high MS resulting from tephra rather than erosion. We did, however, include SARs to help capture this signal. We have clarified which proxies were included in the statistical analyses and why in the revised text. See L270-271.

**4.2.3 Late Holocene**

- L489: To link erosional patterns and nutrient input to human activity, wouldn't it be necessary to have evidence of anthropogenic activities in the watershed of Lake Torfdalsvatn? While regional human signals might be detected through palynology, fire reconstruction, and other proxies with catchment areas larger than the watershed, erosion and productivity related to nutrient input would likely require direct human impact within the immediate watershed. Therefore, is there any archaeological evidence indicating human activity in the watershed of Lake Torfdalsvatn? It may be more accurate to state that the conclusions from L487 onwards apply primarily to the immediate region around the lake, rather than to all of Iceland. Currently, the conclusion drawn in the paragraph starting at L476 may not be fully supported, and the lake record may not reflect this.

The fecal sterol proxies we mention in this paragraph are currently some of the best independent verifications of human presence and impact on individual lake catchments, but challenging to interpret in current Icelandic records. There is some documentary evidence of

farming in the catchment extending back to 1285 CE, which we can include in the revised manuscript. Note, however, that this is several hundred years later than the erosional signals we see. Without a tool, such as fecal sterols, to verify when humans first appeared and impacted the landscape, the conclusions will never be confirmed. Hence, we end this paragraph emphasizing that independent confirmation of humans in the geologic record is required to confidently attribute changes in the landscape to humans or otherwise. See revisions in L739-752.

- L491ff: Throughout the manuscript, the chronological order and logical flow are well-maintained, which I appreciate. However, this paragraph seems to backtrack in time compared to the previous one. I suggest restructuring the order of the entire Late Holocene section (4.2.3) to maintain the chronological order and logical flow.

  After reviewing this portion of the manuscript, we realized there was a typo. The paragraph on the tephra layer disturbance should read "Following the initiation of increased long-term erosion that began around **880 cal a BP**". This has been corrected. See L754.

- One last crucial aspect that is missing is the discussion of the topmost sediments since the end of the Little Ice Age. Was there a re-settlement, was there anthropogenic land use in the vicinity of the lake or in the watershed, and how did this impact the lake? These aspects are vital for anticipating future developments and understanding lake resilience, a key aspect of your study. While erosion (PCoA1) shows an increasing trend towards recent days (2012), algal productivity (PCoA2) does not follow the expected increasing trend, or only does so very weakly. Does this suggest that the anthropogenic signal is weak and that natural variability has much higher amplitudes in this lake? Or does this indicate that, for the period after the LIA, the two PCoAs do not reliably represent the proxies (erosion and algal productivity) anymore?

  We have added some new final sentences at the end of this section highlighting the most recent changes in the proxy records and how they may or may not be consistent with increased human impact on the landscape in the last couple centuries. However, we note that since this sediment core was not measured for radiogenic chronostratigraphic markers, such as [137]Cs, there is some uncertainty in the timing of these proxies at the decadal timescale over the last century. See L793-800.

**5 Conclusions**

- The conclusions are well-supported by the data and results.

- The conclusion about the human settlement around 800 cal BP seems valid. However, what about the modern increase in erosion? Could this be attributed to contemporary land use?

  As we lack sufficient data (chronology and data points) to draw strong conclusions about recent changes over the last couple centuries, we omit any reference to this in our formal conclusion chapter.

**Supplementary data**

- The supplementatry material is a comprehensive and clear tephrography, which is crucial if the paper aims to become a reference stratigraphy. However, I noticed the data tables (SDT1-3) were missing. It is essential to publish all this data so that future studies, not only in Iceland but also globally, can use these as tephra libraries for comparing their tephra layers.

  We apologize for this oversight. The supporting data files will be included in the revision and for the final publication.

**C. Specific comments**

**Minor comments – text**

All minor text comments have been corrected/edited following the reviewer's suggestions.

L57: replace "optimal archive"

L57 ff: break down long sentence.

L63: replace ≤ with ~

L99: JXA-8230 election microprobe à JXA-8230 electron microprobe, replace "l" by "r"

L110: figure reference missing after "Katla 1270"

L139: duplicated occurrence of "from"

L196: with the lowest PSV tie point assigned (to) an age

L198: change "our age model" to non-possessive wording

L198: remove "high-quality" at the end of line

L218: records -> singular

L224: change anticorrelated to "inversely correlated", or use different description of the trends, as no correlation analysis was applied

L237: total chlorin -> plural "chlorins"

L240: abundant pigment -> plural "pigments"

L245: change anticorrelated to "inversely correlated", or use different description of the trends, as no correlation analysis was applied

L287: "[…] due a combination […]": unclear formulation.

L319: refer to Table 2 instead of Table 1

L326: refer to Table 2 instead of Table 1

L386: change anticorrelated to "inversely correlated", or use different description of the trends, as no correlation analysis was applied

L387: "increasingly more abundant", remove "increasingly"

L427: "[…] during first two millenia […]" à "[…] during the first two millenia […]"

L458: between 7.5 and 5.0 cal a BP, Fig. 5 indicates this period rather to be between 7.5 and 5.5 cal a BP

L460: "it is logical" à please reformulate this phrase, and probably the entire sentence starting in L459ff.

L476: indicates à indicate (make it singular, or the "results" plural)

L512: "around 450 BP" -> add cal a BP to be consistent throughout the manuscript.

L526: is it the longest known terrestrial record in Iceland? Or the longest known lacustrine record?

L534: un-identified à unidentified?

**Minor comments – Figures**

All minor figure comments have been corrected/edited following the reviewer's suggestions, except for questions, which are individually address below.

**Figure 1**

- Add the coring location of MD99-2272.
- Caption: Include a comment to highlight the color scheme of the major volcanic systems.

**Figure 2**

- Revise the mean age sample at 3530 cal BP to match Table 1, which lists it as 3520 cal BP.
- Suggestion: While the figure effectively depicts the different tephra layers, it is challenging to correlate them with the data provided in Table 2. I suggest adding the tephra IDs from Table 2 to the figure for better clarity.

**Figure 3**

- Increase the size of panel c to facilitate reading the final age model. Consider moving panels a and b to the supplementary data and enlarging panel c, ensuring that the final age model used is clearly visible, as it is the most important feature of this figure. To facilitate visual discrimination of the different chronomarkers (aquatic microfossil, tephra layer, or PSV-marker), use differently colored symbols as age markers.
- For panel c, clarify whether the tephra deposits and the turbidite (referred to as tephra seismite in a discussion comment) were excised and re-inserted for age modeling. Highlight these layers in the figure if this is the case or provide an explanation if it is not.
- Question: Why does the age-depth model appear relatively stable between the chronomarkers, while the SAR is highly variable? Is it due to the model being "squeezed" in the current figure?
- Caption: Clarify whether "light gray solid lines" refer to dotted lines or if the solid lines are missing. Update the caption or figure accordingly. This may become obsolete if panels a and b are moved to the supplementary data.

**Figure 4**

- Add uncertainty ribbons or ranges to proxies where possible.
- Correction: Correct the plot numbering by adding the missing "c" and removing the duplicate "d".

- Caption: Correct the caption references: "g" should refer to PCoA, and "h" should refer to TLF, as indicated in the figure.
- Caption: Ensure references to plot (G) / (TLF, H) in the caption are printed in lower case: "g" and "h".
- Review: Thoroughly check the figure references in both the text and the plot.
- Suggestion: Add vertical lines, such as dashed lines, at regular intervals (e.g., every millennium) to facilitate readability of the years.

**Figure 5**

- Add uncertainty ribbons or ranges to proxies where possible.
- Suggestion: Add vertical lines, such as dashed lines, at regular intervals (e.g., every millennium) to facilitate readability of the years.
- Suggestion: To make panel c easier to interpret, consider adding an arrow pointing towards areas of higher or lower diatom abundance.
- Question: In panel d, is the high value in the early Holocene (around 10k cal BP) an outlier? Please provide a possible explanation for this high value.

**Figure 7**

- Add uncertainty ribbons or ranges to proxies where possible.
- Suggestion: Add vertical lines, such as dashed lines, at regular intervals (e.g., every millennium) to facilitate readability of the years.
- Correction: Update the y-axis label in panel e from "MD99-2269" to a more appropriate term, such as "sea-ice proxies" or simply "IP25," as the plot includes core MD99-2272 too.

**Figure 8**

- Caption: Revise the text to read: "on the right is (a) simplified schematic…"

**Minor comments – Tables**

All minor table comments have been corrected/edited following the reviewer's suggestions.

**Table 2**

- Add a column containing the core-ID to facilitate finding the tephra layers in Figure 2.

**REVIEWER 2**

**Summary:**

This study is a detailed exploration of the influences of past climate, volcanic eruptions (and resultant tephra) and human settlement on Torfdalsvatn lake, in northern Iceland over the past 12,000 years.

Harning et al. focus on three different datasets: a detailed tephrochronology, bulk physical/geochemical properties and algal pigments. Their tephrochronology work identifies important new regional marker horizons and helps in developing the age model for their Torfdalsvatn sediment record. Bulk geochemical and physical analyses are used to develop a record of catchment stability and soil erosion, both climate driven and anthropogenically disturbed. Finally, algal pigments are used to assess Holocene productivity and the effects of regional climate perturbations. In addition to their data, Harning et al. reference the work that has been previously conducted on sediment cores from this lake and other research in the region.

This paper is an impressive effort to present a substantial dataset and is within the scope of Climate of the Past. The paper is well organized, with figures that are clear and well-presented. Conclusions are well supported by the data presented. The supporting information provides detailed additional data that will be relevant to those interested in the tephrochronology.

Although I am not an expert in either tephra or pigments, I think this is a meaningful contribution to our understanding of Holocene environmental change in Iceland and regional tephrostratigraphy. Although I made multiple comments (mostly minor, with a few more substantial comments) that likely sum up to a major revision, I recommend publication after addressing these points.

We kindly thank Reviewer 2 for their time and consideration reviewing our manuscript that contribute to improving its clarity and focus. Below we address each individual comment in blue and identify where the associated edits have been made.

**Substantial comments:**

- The tephrochronology work in this paper represents a significant dataset (and important contribution) presented in the paper, yet it is not highlighted in the title or introduction. I would recommend both 1) including a reference to the tephrochronology in the title and 2) expanding the introduction to include a paragraph that highlights the importance of tephrochronology related to what this paper has to offer. The first paragraph in the discussion (L275-291) provides nice introductory material about tephras in Iceland, and could be moved to the introduction.

  We have revised the title and introduction to highlight this accordingly. See new title and L68-84.

- The introduction (as currently written) highlights the importance of paleoclimate records for our understanding of anthropogenic warming. The first paragraph is dedicated to this topic. Yet the implications of this new Torfsdalvatn record on our understanding of anthropogenic warming are not specifically highlighted in the discussion/conclusions. In fact, in the discussion the most recent 150 years of the record are not discussed in detail. I recommend exploring this part of the dataset in the discussion and whether there are direct implications from anthropogenic warming recorded in this sediment core. If there is no direct evidence in the sediment core (or even if there is), then what are the implications of this paleo-sediment record for our understanding of anthropogenic warming? I think the discussion could benefit from a more direct approach to this question. Alternatively (or additionally), it may be worth adjusting the first sentence of the introduction to more closely tie to the implications and importance of this particular sediment record.

  We have added new discussion that highlights the most recent changes in the proxy records and how they may or may not be consistent with increased human impact on the landscape in the last couple centuries. See L793-800.

- A big part of the study is refining the ages of tephras and correlating them among multiple sites. The tephras have been dated at multiple sites and thereby provide a rare opportunity to test

the actual accuracy (reproducibility) of lake sediment age models -- not just a statistical estimate of uncertainty output by age modeling algorithms. I'd love to know how well the available ages of the tephras at different sites agree among themselves and with this new reference sequence. What conclusions can be drawn about the veracity of 14C age models from any single lake? Do the reported age model uncertainties hold up to the actual uncertainty based on the timelines set by the tephras?

While we indeed agree that it is important, testing the reliability of age models more generally falls outside the scope of our already dense manuscript. We plan to follow up with this soon in our future research.

• This study builds on previous work on sediment cores from Torfdalsvatn. It would be useful to include a clear statement about how the main findings of this study compare with the previous interpretations, especially the paleoclimate implications. For example, authors of this paper previously reported that Torfdalsvatn is among the lakes that show evidence of an "abrupt summer temperature and catchment disturbance at ~5 ka, statistically indistinguishable from the global 4.2 ka event" (Geirsdóttir et al., 2019; https://doi.org/10.5194/cp-15-25-2019). I didn't see this previously reported major 4.5 ka (= 4.2 ka?) shift mentioned in this study. Do the new data shed new light on previous interpretations?

The impact of ashfall from the Hekla 4 eruption at ~4200 cal yr BP on catchment stability is important to consider at this time, which we do acknowledge in our current discussion. See L643-659. Separating the influence of climate and volcanism on Icelandic sedimentary proxies is a challenge but a current focus of ongoing research.

• I suggest including a figure with the PCoA results, perhaps in the supplement. These results are important to the study, in terms of both erosional activity and algal productivity. The % variance explained by Axis 1 is shown in Figs 4 and 5, but it would be nice for interested readers to have a visual for how these interpretations of axis 1 are derived. I'm thinking of a bivariate PC1/PC2 plot the includes each of the samples.

We have included new PCoA biplots in the revised supplement. See new figures S5 and S6.

• The Supporting Information mentions three supplemental tables, but these are not included in the document. Additionally, I'd also suggest that the data be deposited in a designated repository for paleo data, such as NOAA Paleoclimatology.

The supporting data files will be included in the revision and for the final publication. All proxy data are now archived in NOAA's NCEI database (https://www.ncei.noaa.gov/access/paleo-search/study/39580).

**Minor line comments:**

All minor line comments have been corrected/edited following the reviewer's suggestions.

L14: it's a little confusing to say it's the longest known record but that it is ≤. Consider ~ instead. Same in L63.

L30: consider a first sentence more specific to the importance of this study

L52: sustained is misspelled

L63: be specific rather than using "its". You could say "…, the lacustrine record from Torfdalsvatn has attracted…"

L90: you might consider mentioning that 2012NC was collected from the same coring site as 2012BC, since both are included in the Fig. 1 map

L131-132: consider adding a brief explanation for why terrestrial macrofossils are typically too old, in addition to citing the papers you already do here

L134: add citation for which acid-base-acid pretreatment was used

L139-140: this information is repeated here and in results (L194-196). You might consider saving the details (n = xxx) for the results.

L143: here you say the red line represents mean value of model iterations, but in the caption for Fig 2 you say it's the median value, be consistent.

L153-156: important details about the sampling procedure are missing. What was the sampling resolution (spacing and thickness)? Were samples taken to avoid the tephra layers? Were they sieved to remove macrofossils? Were samples prepared for these analyses using a specific protocol? If yes, cite or describe.

L159: relative abundance of BSi is entirely sufficient for the purposes of this study. Can you say whether absorbance units scale linearly with BSi concentration? I'm curious as to why the values are not calibrated to concentrations (e.g., wt % $SiO_2$).

L165: change $N^2$ to $N_2$ (superscript to subscript 2)

L166: what was sampling resolution?

L217 and L 236: change "equates to 1" to "equates to an average of 1"

L219: perhaps it is worth mentioning the peaks in MS that punctuate the low and stable MS

L219: the mean shift in TOC that occurs ~8ka could be worth mentioning here

L221: replace "be" with "by"

L230-231: for consistency, change to lowercase when referring to figure panels

L242: would "rapidly decrease around 9200 cal a BP" be more accurate?

L245: if saying these are anticorrelated, should provide correlation statistics

L257: replace "present" with "presented"

L294: could streamline text and remove this sentence, which is implied by the previous sentence

L297: continue with format in paragraph and include (xx %) here

L305: 5 regional lake records are shown in Fig 6

L307: change "high" to something like "numerous" or "extensive"

L311: consider these various other factors in more detail for the Torfdalsvatn record. Simple explanation is nice for the regional, but could any of these other factors have influenced the Torfdalsvatn record?

L340: add $\delta^{13}$C in parentheses with C/N

L361: state the timing of deglaciation in the region based on published independent evidence

L392: remove "additional" which will help smooth the transition here

L401: it seems that if low resolution was the explanation, that would show in all of the sedDNA records. Perhaps include an alternative explanation.

L415-420: the transition wording in these sentences is confusing. Clarify if you are saying that the Katla 6500 tephra did impact Torfdalsvatn terrestrial landscape but the Hekla 4 did not.

L430-431: wording here is contradictory, remove "one possibility is that"

L432: add "of" to "both of these proxies"

L438: replace "reads" with "records"

L538: specify the time periods for climate cooling

L552: do you mean "more resistant"

**Minor Figure/Table comments:**

All minor figure/table comments have been corrected/edited following the reviewer's suggestions.

Figure 1:

- In the caption, mention that the colors represent major volcanic systems as indicated in the key.
- Since all other records are symbolized by green (terrestrial) and blue (marine), consider highlighting that Torfdalsvatn, a terrestrial record, is highlighted in yellow

Figure 2:

- The 1310 and 3530 radiocarbon ages do not match the ages presented in Table 1, check this.
- It is slightly confusing to say mean 14C age, but the units are in calibrated age. Consider changing to "mean age (cal yr BP)" to avoid confusion.

Figure 3:

- The color scheme/ key for this figure could be misleading. Grey is used for GREENICE in panels a/b, but is also used for the Torfdalsvatn record in panel d. Consider changing panel d to black for consistency.
- Caption says red line represents median age, but L143 says it represents mean.

Figure 4:

- Change panel label for C/N from "d" to "c".

Figure 8:

- It is difficult to distinguish the dispersed pumice grains from the pumice rich horizons. This may not be highly important to distinguish, so you could consider lumping these together.

---

## Referee Report (RR1)

**Referee Report: Revised Manuscript, Reviewer 2**

1) Scientific significance: excellent (1)
2) Scientific quality: excellent (1)
3) Presentation quality: excellent (1)

For final publication, the manuscript should be: accepted subject to **technical corrections**

**Summary:**
The current version of the manuscript and the response to reviewers document nicely address my comments on the original manuscript. The figures are excellent. Below are a few follow-up minor comments (line numbers based on those in the track changes document). I support accepting this paper after these technical corrections are made.

Thank you for the opportunity to review this work.

**Minor Line Comments:**
L19: consider listing pigments of interest in parentheses
L32: consider adding "for paleoclimate reconstructions" or "for these studies" to clarify why Iceland is an "ideal location"
L105: a key template for what? Being specific here would be good.
L310: add "are" to "are inversely correlated"
L317: "be" should be "by"
L564: "is" should be "are"
L572: consider including the correlation statistics (p, $R^2$)
L766: I think you mean "concomitant" rise in Iceland's population instead of "contaminant"?

**Minor Figure Comments:**
**Figure 1**: it is unclear what the yellow dotted/dashed line indicates, please specify in caption or legend
**Figure 2**: in the figure legend and caption, are the calibrated ages mean or median? They both indicate mean, but I believe they might be median based upon Table 1.
**Figure 4**:
- it looks like the highest value in panel f got cut off around 10,000 cal a BP
- TLF is explained as tephra layer frequency, but it is unclear what exactly this is in relation to. Are the units per year? Per hundred years? Same comment for Figure 5

**Figure S5 and S6**: consider adding the PCoA loadings of different variables onto these plots so that readers can evaluate where datapoints lie in relation to different variables.

---

## Referee Report (RR2)

**Review for Harning et al.** "High-resolution Holocene record based on detailed tephrochronology from Torfdalsvatn, north Iceland, reveals natural and anthropogenic impacts on terrestrial and aquatic environments"

**Structure of review:** A. Summary, B. Minor comments

**A. Summary**

Harning et al. have effectively incorporated most of the reviewers' feedback, thoroughly addressing and responding to all suggestions. Notably, the title, abstract, and introduction now appropriately incorporate tephrostratigraphy, enhancing the manuscript's thematic coherence. The methods section has also been reorganized, presenting a clearer and more structured approach than in the initial version.

Despite the density of the manuscript, which still presents extensive datasets that could constitute two independent papers, I found it engaging and insightful.

I am pleased to recommend it for publication, pending the authors' addressing of some minor comments.

**B. Minor Comments**

**1 Introduction**

L34: "…have analyzed physical…" Have the empirical records analyzed the properties?

L57: "Icelandic lake sediment studies provide optimal archives…" do the studies provide archive, or the sediments?

**2 Methods**

From an organizational perspective, the manuscript is now concisely structured.

**3 Results**

L237: As your Fig. 3b shows, the SAR is not linear, but you have variations from 0.001-0.25, which is >20-fold of variation. This substantial variability could influence the interpretation of sedimentary fluxes and patterns.
My question about MAR from the first round has been replied to as follows: "Due to the linear sediment rate, calculating fluxes does not alter the structure of the proxy curves and therefore does not provide any difference in plots than those shown in the figures presented. This point was already made in the original manuscript. See L300." While this explanation is understandable, it overlooks the considerable variability in SAR noted in Fig. 3b. MAR (mass accumulation rate, expressed as mass per area and year, e.g., g cm$^{-2}$ yr$^{-1}$) differs from SAR (sediment accumulation rate, e.g., cm yr$^{-1}$). Typically, substance fluxes are calculated using MARs. Given the changes in the sedimentary matrix (minerogenic versus biogenic, as you have nicely detailed in your manuscript) and the variations in $C_{org}$, alongside changes in SAR, it is reasonable to anticipate a different flux pattern compared to the concentrations. I suggest calculating these fluxes and presenting the results in the supplementary information. If they are substantially different (in terms of trends), it would be interesting to see whether they would change the interpretation of the dataset.

**4 Discussion**

The first paragraph (L316–327) primarily describes and quantifies the tephra layers in a descriptive manner. To enhance the flow and logical structure of the manuscript, I recommend moving this section to the results section. Consequently, the first paragraph of the discussion (starting at L328) should be adjusted to reflect this change and maintain coherence.

L323: I couldn't find the Veidivötn-B in Table 2 (also comment later) → make sure to be consistent with IDs

L370: "have low C/N (7 to 26) and high d13C values..." → add "relatively" → "relatively low" and "relatively high" values

L379: "...as they reflect all taxonomic groups..." remove "all" → ...as they reflect taxonomic groups...

L392: in front of 10,000 is there a double spacing? Use search&find function in word processor to find and remove double spacing.

L479: are there other confounding factors influencing the abundance of cyanos, e.g. anoxic periods?

L568: "the contaminant rise..." is this speculative? Rephrase it to anthropogenic landuse, or pressure or similar.

**5 Conclusions**

L573: "high-resolution age model" is overselling here. Varves would be high resolution, or age control every 100 years, here it is every 550 years.

BTW. In Figure 2 I only count 18 age markers, but you claim that there are 22 in the conclusion, and in the figure legend you describe 20, what is correct? → be consistent

**Supplementary data**

The supplementary material provides a comprehensive and well-structured tephrography, which is clear and informative. I also appreciate that the data tables are now accessible. However, I noticed that the cluster coordinates for the tephra layers (e.g., 'Hekla') are not provided in a tabular format. Including these coordinates in a spreadsheet would enhance the utility of the supplementary data for future users.

Aside from this, the collection of biplots is exhaustive and very well done—great work!

**Minor comments - Figures**

**Figure 3**

- How many control points? I count 18 in the figure, see 20 in the figure caption, and 22 in the conclusion section...

**Figure 4**

- The PCoA Axis 1 curve (plot f) seems to be cut-off in the Early Holocene, is this on purpose?
- Suggestion: Add vertical lines, such as dashed lines, at regular intervals (e.g., every millennium) to facilitate readability of the years.

**Figure 8**

- Increase scale bar (on the left of the figure), it is not readable as is

**Minor comments - Tables**

**Table 2**

- Action: Add a column containing the core-ID to facilitate finding the tephra layers in Figure 2. This hasn't been implemented yet.
- In the footer of the table you describe "V-B, Veidivötn...", however, there is no Tephra ID visible in the table, either remove this, or add the tephra layer.

---

## Author Response (AR2)

**REVIEWER 1**

**Review for Harning et al.** "High-resolution Holocene record based on detailed tephrochronology from Torfdalsvatn, north Iceland, reveals natural and anthropogenic impacts on terrestrial and aquatic environments"

**Structure of review:** A. Summary, B. Minor comments

**A. Summary**

Harning et al. have effectively incorporated most of the reviewers' feedback, thoroughly addressing and responding to all suggestions. Notably, the title, abstract, and introduction now appropriately incorporate tephrostratigraphy, enhancing the manuscript's thematic coherence. The methods section has also been reorganized, presenting a clearer and more structured approach than in the initial version.

Despite the density of the manuscript, which still presents extensive datasets that could constitute two independent papers, I found it engaging and insightful.

I am pleased to recommend it for publication, pending the authors' addressing of some minor comments.

We kindly thank the reviewer for their continued time and effort providing detailed suggestions for manuscript improvement. Below we address each of their comments in blue.

For what it's worth, we also agree this could be two papers! However, a previous attempt to publish this as paired manuscripts (tephra and paleoclimate) was unsuccessful with reviewers suggesting it be merged into one…

**B. Minor Comments**

**1 Introduction**

L34: "...have analyzed physical..." Have the empirical records analyzed the properties?

Changed to "include"

L57: "Icelandic lake sediment studies provide optimal archives..." do the studies provide archive, or the sediments?

Removed "studies"

**2 Methods**

From an organizational perspective, the manuscript is now concisely structured.

Thank you for the feedback.

**3 Results**

L237: As your Fig. 3b shows, the SAR is not linear, but you have variations from 0.001-0.25, which is >20- fold of variation. This substantial variability could influence the interpretation of sedimentary fluxes and patterns.

My question about MAR from the first round has been replied to as follows: "Due to the linear sediment rate, calculating fluxes does not alter the structure of the proxy curves and therefore does not provide any difference in plots than those shown in the figures presented. This point was already made in the original manuscript. See L300." While this explanation is understandable, it overlooks the considerable variability in SAR noted in Fig. 3b. MAR (mass accumulation rate, expressed as mass per area and year, e.g., g $cm^{-2}$ $yr^{-1}$) differs from SAR (sediment accumulation rate, e.g., cm $yr^{-1}$). Typically, substance fluxes are calculated using MARs. Given the changes in the sedimentary matrix (minerogenic versus biogenic, as you have nicely detailed in your manuscript) and the variations in $C_{org}$, alongside changes in SAR, it is reasonable to anticipate a different flux pattern compared to the concentrations. I suggest calculating these fluxes and presenting the results in the supplementary information. If they are substantially different (in terms of trends), it would be interesting to see whether they would change the interpretation of the dataset.

We apologize for the confusion from the initial round of reviews. We had mistakenly calculated MARs without incorporating density into the equation. We have now added MARs for C, BSi, and algal pigments in the supplementary information for the reader's reference (Fig. S5) and reference them in the results although we retain our discussion focus on the relative changes and concentrations.

**4 Discussion**

The first paragraph (L316–327) primarily describes and quantifies the tephra layers in a descriptive manner. To enhance the flow and logical structure of the manuscript, I recommend moving this section to the results section. Consequently, the first paragraph of the discussion (starting at L328) should be adjusted to reflect this change and maintain coherence.

Thank you for the suggestion. We have moved this paragraph to the Results Section 3.1 – Tephra stratigraphy and age model.

L323: I couldn't find the Veidivötn-B in Table 2 (also comment later)◊make sure to be consistent with IDs

Veiðivötn-Barðarbunga tephra shards are present in some of the mixed tephra layers, as denoted in the second column of Table 2. As no tephra layers are dominantly from the V-B volcanic system, we have removed the V-B footnote from Table 2. We hope this clarifies the issue. However, in L323, we do provide reference to the supplemental material where these V-B tephra shards are described in detail.

L370: "have low C/N (7 to 26) and high d13C values..."◊add "relatively"◊"relatively low" and "relatively high" values

Edited

L379: "...as they reflect all taxonomic groups..." remove "all" ◊ ...as they reflect taxonomic groups...

Edited

L392: in front of 10,000 is there a double spacing? Use search&find function in word processor to find and remove double spacing.

We searched and found no double spaces.

L479: are there other confounding factors influencing the abundance of cyanos, e.g. anoxic periods?

We do not currently have information on past anoxia from this lake but it is the motivation of some pending work.

L568: "the contaminant rise..." is this speculative? Rephrase it to anthropogenic landuse, or pressure or similar.

Contaminant was an error – meant to read "concomitant". This has been corrected and aligns with the reviewer's suggestion.

**5 Conclusions**

L573: "high-resolution age model" is overselling here. Varves would be high resolution, or age control every 100 years, here it is every 550 years.

High-resolution edited to "detailed".

BTW. In Figure 2 I only count 18 age markers, but you claim that there are 22 in the conclusion, and in the figure legend you describe 20, what is correct?◊be consistent

Apologies for the confusion. There are 20 age control points. 22 was mistakenly written and two PSV tie points were accidentally not marked as they overlapped with 14C ages. These issues have now been fixed.

**Supplementary data**

The supplementary material provides a comprehensive and well-structured tephrography, which is clear and informative. I also appreciate that the data tables are now accessible. However, I noticed that the cluster coordinates for the tephra layers (e.g., 'Hekla') are not provided in a tabular format. Including these coordinates in a spreadsheet would enhance the utility of the supplementary data for future users.

Aside from this, the collection of biplots is exhaustive and very well done—great work!

The cluster coordinates are usable for any potential user through the plots provided in the supplement. We are still working to publish this database as a formal study that will include such data so that users can plot volcanic source fields in their preferred software.

Thank you for the feedback!

**Minor comments - Figures**

**Figure 3**

- How many control points? I count 18 in the figure, see 20 in the figure caption, and 22 in the conclusion section...

  20 – see reply above

**Figure 4**

- The PCoA Axis 1 curve (plot f) seems to be cut-off in the Early Holocene, is this on purpose?

  We apologize for the error cutting of the data slightly. The y-axis has been expanded to include the Early Holocene data.

- Suggestion: Add vertical lines, such as dashed lines, at regular intervals (e.g., every millennium) to facilitate readability of the years.

  We tried this suggestion during the first round of revision and found it made the plots too busy with the other bars and guided annotations. As such, we prefer to not add these to this paper.

**Figure 8**

- Increase scale bar (on the left of the figure), it is not readable as is

  Edited

**Minor comments - Tables**

**Table2**

- Action: Add a column containing the core-ID to facilitate finding the tephra layers in Figure 2. This hasn't been implemented yet.

  Only two tephra layers from Table 2 are used in the age model (H 1766 and H 1300). To facilitate easier connection between the two, we have removed abbreviations in Table 2 for tephra names.

- In the footer of the table you describe "V-B, Veidivötn...", however, there is no Tephra ID visible in the table, either remove this, or add the tephra layer.

  Following an earlier related comment we have removed V-B from the footnote.

**REVIEWER 2**

1. 1)  Scientific significance: excellent (1)
2. 2)  Scientific quality: excellent (1)
3. 3)  Presentation quality: excellent (1)

For final publication, the manuscript should be: accepted subject to **technical corrections**

**Summary:**

The current version of the manuscript and the response to reviewers document nicely address my comments on the original manuscript. The figures are excellent. Below are a few follow-up minor comments (line numbers based on those in the track changes document). I support accepting this paper after these technical corrections are made.

Thank you for the opportunity to review this work.

We kindly thank the reviewer for their continued time and effort providing detailed suggestions for manuscript improvement. Below we address each of their comments in blue.

**Minor Line Comments:**

L19: consider listing pigments of interest in parentheses
L32: consider adding "for paleoclimate reconstructions" or "for these studies" to clarify why Iceland is an "ideal location"
L105: a key template for what? Being specific here would be good.
L310: add "are" to "are inversely correlated"
L317: "be" should be "by"
L564: "is" should be "are"

L572: consider including the correlation statistics (p, $R^2$)
L766: I think you mean "concomitant" rise in Iceland's population instead of "contaminant"?

All line comments edited

**Minor Figure Comments:**
**Figure 1**: it is unclear what the yellow dotted/dashed line indicates, please specify in caption or legend

The yellow dotted lines were remnant from an older version of the manuscript that is not discussed here. We have removed them now.

**Figure 2**: in the figure legend and caption, are the calibrated ages mean or median? They both indicate mean, but I believe they might be median based upon Table 1.

14C ages are presented as mean ± 1σ as written

**Figure 4**:

- it looks like the highest value in panel f got cut off around 10,000 cal a BP

  We apologize for the error cutting of the data slightly. The y-axis has been expanded to include the Early Holocene data.

- TLF is explained as tephra layer frequency, but it is unclear what exactly this is in relation to. Are the units per year? Per hundred years? Same comment for Figure 5

  TLF is the number of events identified in each tephra layer (see Table 2). We have now clarified this in captions for Figs 4 and 5.

**Figure S5 and S6**: consider adding the PCoA loadings of different variables onto these plots so that readers can evaluate where datapoints lie in relation to different variables.

Added